



# Decomposing the Tea Bag Index and finding slower organic matter loss rates at higher elevations and deeper soil horizons in a minerogenic salt marsh.

Satyatejas G. Reddy[1], W. Reilly Farrell[1], Fengrun Wu[2], Steven C. Pennings[3], Jonathan Sanderman[4], Meagan Eagle[5], Christopher Craft[6], Amanda C. Spivak[1]*

[1]University of Georgia, Marine Sciences Department, Athens, GA, USA 30602

[2]Xiamen University of Technology, School of Environmental Science and Engineering, Xiamen,
Fujian Province, China

[3]University of Houston, Department of Biology and Biochemistry, Houston TX 77204, USA

[4]Woodwell Climate Research Center, Falmouth, MA, USA

[5]United States Geological Survey, Marine Science Center, Woods Hole, MA, USA

[6]Indiana University Bloomington, School of Public and Environmental Affairs, Bloomington IN
47405, USA

*corresponding author: aspivak@uga.edu

Keywords: decay, decomposition, salt marsh, organic matter, tea bag index, hydrology

Running title: TBI and organic matter decay in marshes





**Abstract**

Environmental gradients can affect organic matter decay within and across wetlands and contribute to spatial heterogeneity in soil carbon stocks. We tested the sensitivity of decay rates to tidal flooding and soil depth in a minerogenic salt marsh using the tea bag index (TBI). Tea bags were buried at 10- and 50- cm along transects sited at lower, middle, and higher elevations that

paralleled a headward eroding tidal creek. Plant and animal communities and soil properties were characterized once while replicate tea bags and porewaters were collected several times over one year. TBI decay rates were faster than prior litterbag studies in the same marsh, largely due to rapid green tea loss. Rooibos decay rates were comparable to natural marsh litter, potentially suggesting that is more useful as a standardized organic matter proxy than green tea. Decay was slowest at

higher marsh elevations and not consistently related to other biotic (e.g., plants, crab burrows) and abiotic factors (e.g., porewater chemistry), indicating that local hydrology strongly affects organic matter loss rates. Tea BI rates were 32-118% faster in the 10 cm horizon compared to 50 cm. Rates were fastest in the first three months and slowed 54-60% at both depths between 3- and 6- months. Rates slowed further between 6- and 12- months but this was less dramatic at 10 cm (17%)

compared to 50 cm (50%). Slower rates at depth and with time were unlikely due to the TBI stabilization factor, which was similar across depths and decreased from 6 to 12 months. Slower decay at 50 cm demonstrates that rates were constrained by the environmental conditions of this deeper horizon rather than the molecular composition of litter. Overall, these patterns suggest that hydrologic setting, which affects oxidant introduction and reactant removal and is often

overlooked in marsh decomposition studies, may be a particularly important control on organic matter decay in the short term (3 – 12 months). transects sited at lower, middle, and higher elevations that paralleled a headward eroding tidal creek.





## 1. Introduction

Long term sustainability of salt marshes and their role as a carbon sink depend on efficient
preservation of organic matter. Preservation is generally ascribed to a combination of rapid
deposition of refractory organic matter and slow decay in anoxic soils (Benner et al., 1991; Morris
& Bowden, 1986; Valiela et al., 1985). Yet, variability in soil carbon stocks and accumulation
rates within and across marshes argues that controls on preservation are more complex (Arriola &
Cable, 2017; Holmquist et al., 2018). Differential tidal flooding across marsh elevations affects
aboveground plant production and the belowground soil environment that, in turn, constrains
microbial access to oxidants and organic matter (Guimond & Tamborski, 2021; Morris & Bowden,
1986; Spivak et al., 2019). In surface horizons, marsh plant roots and animal burrows further alter
soil structure and chemical gradients that affect organic matter decay, while deeper horizons are
generally more stable environments (Gribsholt & Kristensen, 2002; Guimond et al., 2020).
Characterizing patterns in organic matter decay across tidal inundation gradients and soil depths
may therefore provide a useful framework to assess processes contributing to marsh-scale spatial
variability in carbon stocks.

Tidal hydrology structures salt marsh landscapes and influences productivity, but effects
on soil organic matter decay are less well understood. Flooding changes the soil environment for
decomposition by leaching organic matter, altering availability of terminal electron acceptors,
increasing pore space connectivity, and changing microbial access to bioavailable compounds
(e.g., sorption, enzyme functionality, molecular configuration) (Bradley & Morris, 1990; Giblin &
Howarth, 1984; Liu & Lee, 2006; Morrissey et al., 2014). Soil redox conditions change with rising
and falling tides, indicating that water passage through the subsurface alters the thermodynamic
favorability of different pathways for decomposition (Fettrow et al., 2023; Seyfferth et al., 2020;
Spivak et al., 2023). The intensity of tidal flooding effects on plant and soil processes is strongest
at creekbanks and lower elevations relative to interior and higher elevation areas (Guimond &
Tamborski, 2021; Howes & Goehringer, 1994; Reed & Cahoon, 1992).

Plant and animal effects on soil decomposition are layered on top of this hydrogeomorphic
template. The classic parabolic relationship between flooding and plant productivity suggests there
is an optimal flooding regime, where productivity and, presumably, root release of oxygen and
bioavailable carbon compounds (hereafter, exudates) peaks, with elevations above and below
receiving fewer inputs into the rhizosphere to fuel soil microbes (Megonigal et al., 1999; Mueller



et al., 2016; Spivak & Reeve, 2015). The physical properties of surficial soils are also altered by
roots and animal burrows which reduce bulk density and allow for greater infiltration of tidal
waters that deliver oxygen to anoxic horizons. Soils buried below animal burrows and the active
rooting zone are more insulated from inputs of oxygen and exudates and daily tidal oscillations,
typically have higher bulk densities, and are more compacted (Turner et al., 2006). This more
stable soil environment is likely a key reason that decomposition slows with depth, but attributing
causality is complicated by differences in organic matter composition between surface and deeper
horizons (Bulseco et al., 2020; Luk et al., 2021; Yousefi Lalimi et al., 2018). Standardized litter
approaches offer ways to test controls on organic matter loss across ecological, geomorphic, and
spatial gradients while avoiding potential confounding factors of litter composition and
preparation.

85          Decades of field, lab, and theoretical experiments report a wide range of decay rates, but
robust relationships with climatic, landscape, and ecological drivers remain difficult to quantify
(Charles & Dukes, 2009; Kirwan et al., 2013, 2014; Kirwan & Blum, 2011; Mueller et al., 2016;
Noyce et al., 2023; Tang et al., 2023). The diversity of decay rates likely reflects spatial
heterogeneity within wetlands as well as the variety of measurement techniques applied over
different time scales (Blum, 1993; Kirby & Gosselink, 1976; Luk et al., 2023; Luk et al., 2021;
Newell et al., 1989). Litter bags have the advantage of assessing mass loss of local marsh detritus,
but this bulk approach lacks sensitivity and results can be difficult to compare across studies due
to differences in the use of above- and/or below- ground material, deployment duration, and depth,
among other factors (Blum & Christian, 2004; Charles & Dukes, 2009; Christian, 1984; Kirwan
& Blum, 2011; Windham, 2001). Geochemical approaches describe organic matter loss and
transformations, can be applied over timescales of seasons to centuries, and benefit from multiple
proxies, but are resource intensive and require specialized instrumentation (Benner et al. 1984a,
Benner et al. 1984b; Benner et al., 1987; Benner et al., 1991; Duddigan et al., 2020; Luk et al.,
2023; Luk et al., 2021; Moran et al., 1989).

100          The Tea Bag Index (TBI) is an alternative approach that has been widely adopted because
the standardized method and materials allow for greater comparability between studies; it is
inexpensive, does not require specialized instrumentation, and can be accomplished within months
by non-experts (Keuskamp et al., 2013; Mueller et al., 2018). However, like all methods, the TBI
rests on assumptions. One key assumption is that the decay dynamics of two different litter types





(green and rooibos teas) can be integrated to estimate loss of natural detritus that has characteristics of the proxy constituents. The TBI, in effect, assumes that a two-pool decay model and the biochemical composition of both pools are broadly applicable. Chemical characterization of green and rooibos teas during a 91-day incubation described changes in carbon functional groups that are consistent with decomposition and not dissimilar to natural litter, supporting the TBI approach

(Duddigan et al., 2020). However, green tea is rich in tannins which are lost rapidly (Duddigan et al., 2020), raising the question of whether the mechanism is leaching, microbial decomposition, or some combination (Lind et al., 2022).

Here, we aimed to gain insight to spatial and temporal patterns in decomposition by testing how tidal inundation affects organic matter decay rates across soil depths and over time in a salt

marsh. We employed the TBI to examine the effects of these environmental gradients on decay without the potential confounding factor of varying organic matter composition. We hypothesized that slower TBI decay rates would coincide with lower inundation rates and more stable conditions below the rhizosphere. In addition, we tested several key assumptions of the TBI method by extending the prescribed incubation time from 3 to 12 months and comparing the decay dynamics

of rooibos and green teas to prior studies using local plant detritus in the same minerogenic marshes. We considered whether sharp drops in TBI decay rates over short time scales (i.e., 3 months), particularly at shallow soil depths and lower marsh elevations with frequent flooding, could be due to leaching rather than decay, whereas rate differences that persist across depth and elevation gradients for longer periods (i.e., 6- and 12- months) would be more representative of

environmental constraints on decomposition.

## 2. Methods

### 2.1 Study site and design

We tested whether TBI decay rates differ within vs. below the rhizosphere of *Spartina*

*alterniflora* marshes within the Georgia Coastal Ecosystems Long-Term Ecological Research (GCE-LTER) domain (31.421° N, -81.290° W). Tides are semidiurnal with a ~2 m range, and the marsh is dominated by *S. alterniflora*. Study plots were established along a tidal creek, with 8 plots at 3 distances from the creekbank edge (near: 0 m, mid: 4 m, and far: 14 m; resulting in 24 plots; *S. alterniflora* populated all but one plot which was removed from data analyses). Study plots were

situated at elevations ranging from 0.55 to 1.07 m (NAVD 88). Plots farther away from the creek



sat at higher elevations (0.74–1.07 m) than those located next to the creek (0.55-0.77 m). Tea bag decay rates and porewater chemistry were measured at two depths (10 and 50 cm) at discrete intervals over one year (July 2019-2020). Soil temperatures were continuously monitored for ~6 months at both depths. This study was conducted alongside that of Wu et al. (2022), who measured

plot elevations, plant and animal community characteristics, and soil shear strength once during summer 2019, which we use to contextualize and interpret our results.

### 2.2 Marsh Surface Elevation

Marsh surface elevations were measured within 2 m of each plot, to minimize trampling,

using a Trimble R6 Real-Time Kinematic Global Positioning System receiver (Table S1). Elevation data are referenced to the North American Vertical Datum of 1988 (m, NAVD 88). Relative marsh surface elevation of each plot within the tide frame ($Z^*$) was calculated as

$$Z^* = (\text{NAVD88 elevation-MSL}) / (\text{MHHW-MSL}), \qquad (1)$$

where MSL is mean sea level and MHHW is mean highest high water, referenced to the nearest

NOAA tide station (Fort Pulaski, GA 8670870). Elevation data were also used in combination with tide heights, recorded in a nearby creek (31.4437673 °N, -81.2838603 °W), to distinguish between periods of tidal inundation and exposure. Tide heights were recorded in 5-minute intervals by a titanium pressure transducer (Campbell Scientific™ CS456) deployed at a verified elevation and operated by the GCE-LTER project.


### 2.3 Temperature

Soil temperature at 10 and 50 cm depths was recorded by HOBO loggers (UA-002-08, Onset Computer Corp, accuracy: ± 0.53° C from 0° to 50° C) in 15-minute intervals. Loggers were intercalibrated prior to deployment (SE ± 0.07° C). The loggers were deployed in July 2019 and

collected 188 days (~6 months) later in January 2020. We calculated the average, minimum, and maximum daily temperatures for each of the 15 loggers deployed at 10 cm and the 16 loggers deployed at 50 cm that were recovered and functioning.

### 2.4 Porewater Chemistry

One passive porewater sipper was deployed in each plot in July 2019 with collection windows at 10 and 50 cm (Hughes et al 2012, Paludan and Morris 1999). A single glass



scintillation vial, filled with Milli-Q water (18.2 MΩ) and fitted with an open top cap and 50 μm Nitex mesh, was placed upside down in each collection window. Porewater vials were retrieved two months later, reflecting our expectation of dynamic changes during summer, and again at 98,

188, and 363 days, which correspond with the 3- , 6-, and 12- month teabag collections. Collected vials were replaced in the sippers with fresh vials and Milli-Q water. Samples were sealed with solid caps and transported on ice to the University of Georgia Marine Institute where salinity, redox, and pH levels were measured. This sampling approach relies on equilibration of water inside the vial with the surrounding porewater, which happens within one month and was assessed

based on salinity readings. Salinity was measured with a handheld refractometer while pH levels and redox potential were measured with a benchtop dual channel pH/ISE meter (Fisherbrand™ Accumet™ XL250, accuracy ±0.002 pH units) and a calibrated pH combination electrode (Fisherbrand™ accuTupH™) or redox oxidation / reduction potential electrode (Mettler Toledo™ InLab™ Redox ORP Electrode), respectively. Redox potential readings (mV) were recorded

relative to a reference electrode in a 3.5 M potassium chloride solution and values were subsequently corrected to the standard hydrogen electrode.

### 2.5 Plants, animals, and soil stiffness

Plant characteristics, animal abundances, and soil stiffness were reported previously by Wu

et al., 2022 in summer 2019 in the same nearby plots where elevations were measured. Briefly, soil shear strength was measured in the top 4 cm using a field shear vane (GEONOR H-6O) and was lower close to the creek than far away. *Spartina alterniflora* aboveground biomass was estimated based on stem density counts and known masses of representative stems. In general, plant density was lower, but stem height and plant biomass greater close to the creek versus farther

away. Belowground biomass was measured by collecting soil cores (10 cm diameter, 30 cm depth) centered on a culm of *S. alterniflora* in each plot and then washing roots and rhizomes free of soil before drying and weighing. Two major groups of invertebrates were present: crabs (*Uca pugilator*, *Sesarma reticulatum, Panopeus*) and snails (*Littoraria irrorata*). The densities of crab burrows (>0.5 cm diameter, all species pooled) and snails (>0.3 cm spire height) were recorded in

0.5 × 0.5 m quadrats at each plot. The density of crab burrows and snails tended to be greater far from the creek versus close to the creek.



### 2.6 Decay rates

Decay coefficients (hereafter, rates) were approximated by measuring mass loss over time

of a standardized litter. We chose the Tea Bag Index (TBI) because this approach has been used broadly across ecosystem types, allowing for intercomparisons (Keuskamp et al., 2013). This method assumes that natural litter is comprised of labile and refractory pools that turnover at different rates and can be represented by Lipton™ green (European Article Number: 87 22700 05552 5) and rooibos (European Article Number: 87 22700 18843 8) teas, respectively. Teabags

were dried at 60 °C to constant mass and triplicate bags of each tea type were buried in every plot at 10 and 50 cm depth in July 2019. Single replicates were collected after 98 days (~3 months), 188 days (~6 months), and 363 days (~12 months, July 2020). Collected tea bags were again dried at 60 °C and mass loss was calculated as the difference between the dried initial and final tea masses, after correcting for contributions from the tea bag, string, and label.

Decay rates were calculated per Keuskamp et al. (2013) using three equations:

$$W(t) = a_r e^{(-kt)} + (1 - a_r), \qquad (2)$$

$$S = 1 - \frac{a_g}{H_g}, \qquad (3)$$

$$a_r = H_r(1 - S). \qquad (4)$$

The variable $W(t)$ is the mass fraction of rooibos tea remaining at time $t$, $k$ is the decay coefficient

(i.e., rate), and $S$ is a stabilization factor. The tea-specific variables of $a_r$ and $a_g$ are the decomposable fractions, and $H_r$ and $H_g$ are operationally defined hydrolysable ($H$) fractions of rooibos and green teas, respectively. The decomposable fraction of green tea ($a_g$) was calculated as the mass fraction lost over a given time. Kesukamp et al. (2013) calculated $H_g$ and $H_r$ as 0.842 and 0.552 (g g$^{-1}$), respectively. The stabilization factor is meant to represent the conversion of

labile to refractory organic matter and calculated as a deviation from the mass fraction that is decomposed relative to the fraction that is hydrolysable.

We then compared TBI decay rates with tea-specific rates estimated from a first order decay model:

$$a = a_o e^{(-kt)}, \qquad (5)$$

where $a$ is the tea mass fraction remaining after a given amount of time, $t$, $a_o$ is the initial mass fraction (i.e., 1), and $k$ is the calculated decay coefficient. Single exponential models were fitted to tea-specific mass loss fractions at 0, 98, 188, and 363 days and included 2, 3, or 4 time points, respectively, producing $k_g$ and $k_r$ (Fig. S1). The fraction of variance explained by the decay models



was generally greater for green tea than rooibos. We suspect this is because mass loss rates are a

fairly insensitive metric and were much slower for rooibos than green tea. Only models with $r^2$

values $> 0.60$ were included for the 6- and 12- month time points to be as representative of the full

dataset as possible (i.e., 61% and 100% for the rooibos and green tea bags, respectively). Outliers

were removed prior to fitting equation 5 for the 3-month tea bags (see 2.7). To further assess

assumptions of the TBI approach we calculated tea-specific decomposable fractions ($a$) and

stabilization factors ($S$). Decomposable fractions were defined as mass lost during an incubation

and calculated as $1-a$, resulting in $a_g$ (same as Keuskamp et al. 2013) and $a_r$, for green and rooibos

teas, respectively. We used Keuskamp et al.'s formulation of $S$ in equation 3 to represent $S_g$, but

modified it for rooibos tea ($S_r$) by substituting $a_r$ and $H_r$. From here forward, rates and variables

calculated using Keuskamp et al. (2013) or the first order decay approach are referred to TBI and

empirical, respectively.

**2.7 Decay rates from marsh litterbags**

Geochemical changes of natural, marsh organic matter undergoing decomposition have

been well studied in Georgia marshes (Benner et al., 1984; Benner et al. 1987; Benner et al., 1991;

Rice & Tenore, 1981). However, we were unable to find published organic matter mass loss rates

from litterbags, which would be a more comparable complement to the TBI. Instead, we draw on

a prior experiment conducted June 2003-2004. Roots were collected from the levee and plain of a

*S. alterniflora* marsh within GCE-LTER. Decay rates were measured following the methods of

Blum (1993) in which 10 g of root material was placed in nylon mesh (2 mm x 2 mm) bags (30

cm x 7 cm) and buried (10-20 cm) for up to one year. Sixteen replicates each were initially buried

in the marsh levee and plain, and four replicates were retrieved from both sites every ~ 3 months.

Bags were transported on ice to the lab and dried at 70° C to constant mass. Decay rates were

calculated as in equation 5.

**2.8 Data analyses**

Changes in belowground environmental conditions across marsh surface elevations,

between soil depths, and over time were assessed with regression analyses and t-tests. Tidal

flooding effects on soil porewater chemistry and temperature were tested by constructing

regression models against relative elevation (Z*). Porewater data were then aggregated by





sampling event and two sample t-tests were used to detect differences between 10 cm and 50 cm depths. Correlations between Z* and soil temperature were further tested by partitioning according to season (summer: 68 days between 18 July-22 September 2019, fall: 91 days between 23 September -22 December 2019; winter: 28 days between 23 December 2019-19 January 2020) and periods of tidal inundation or exposure; differences between slope coefficients were evaluated

based on Clogg et al., 1995 We then tested whether soil temperatures differed between depths within each season using paired t-tests.

We tested whether TBI and empirical decay rates (k, $k_g$, and $k_r$; $d^{-1}$) and stabilization factors (S, $S_g$, and $S_r$) differed over time (3-, 6-, or 12-months) and between soil depths (10 vs 50 cm) by constructing linear mixed effect models using the nlme package for R (Pinheiro et al., 2016). The

mixed models evaluated the fixed effects of time and depth and included plot number as a random factor. We then conducted paired t-tests to further explore how TBI and empirical decay rates, decomposable fractions ($a_g$, $a_r$), and stabilization factors changed over time within a depth horizon.

Potential drivers of TBI k ($d^{-1}$) were evaluated by calculating Spearman rank correlation coefficients between rates and environmental conditions for the three time points (3, 6, or 12

months) and two soil depths (10 and 50 cm). The analyses included time-point specific porewater chemistry (i.e., redox, pH, and salinity) and contextual data collected in summer 2019 (i.e., animal burrows, snail abundances, plant biomass, soil stiffness, relative elevation (Z*)). Porewater data for the 3, 6, and 12 month periods were combined with data from previous time points (e.g., 2, 3, or 6 months) to better represent cumulative conditions. The rationale was that decay rates reflect

environmental conditions over the entire deployment period. Temperature was excluded because the shared time series with TBI k violated assumptions of independence. The TBI k values correlated strongly with relative elevation (Z*), *S. alterniflora* rhizome and aboveground biomass, and soil stiffness. Because tidal flooding structures salt marsh habitats, we conducted subsequent single-factor regressions of relative elevation (Z*) with aboveground biomass and soil stiffness.

Correlations between these variables limited further hypothesis testing of decay drivers to plot position within the tidal frame (Z*). We tested whether TBI k rates and S values changed with relative elevation (Z*) using linear regression models and then evaluated differences between the resulting slope coefficients over time and with depth, as described by Clogg et al., 1995.

Data were tested for outliers using a 1.5 interquartile range cutoff and transformed as

needed to meet assumptions of normality. Analyses were conducted using R software (R





Development Core Team, 2024. Data are presented as means ± standard error (SE) unless noted otherwise.

## 3. Results

### 3.1 Soil porewater chemistry

Porewater chemistry differed between depths and changed over the year but was surprisingly insensitive to relative marsh elevation ($Z^*$). Salinities were lower at 10 cm than 50 cm after the first 2 months of the experiment (Fig. 1a) and correlated positively with relative elevation ($Z^*$) at 50 cm ($r^2=0.28$, $p<0.05$) but not at 10 cm ($r^2=0.17$, $p>0.05$). Redox potentials

were more oxidizing in the shallower horizon and generally decreased over time at both depths but did not vary with relative elevation ($Z^*$) at 10 cm ($r^2=0.06$, $p>0.05$) or 50 cm ($r^2=0.00$, $p>0.05$) (Fig. 1b). Porewater pH levels were similar between depths, or slightly higher at 10 cm, with little change over the year (Fig. 1c) and no change with relative marsh elevation ($Z^*$) at either 10 cm ($r^2=0.17$, $p>0.05$) or 50 cm ($r^2=0.15$, $p>0.05$).

Temperature differences between 10 and 50 cm were slight (~1 °C) and changed seasonally, with tidal inundation and relative marsh elevation ($Z^*$) (Fig. 2a-f; Table 1). The warmest temperatures were at 10 cm during summer but temperatures were higher at 50 cm in the fall and winter (Table 1A). Soil temperatures were similar across tidal stages in the summer and winter but were warmer during periods of inundation in the fall. Temperatures at 10 cm decreased

with increasing relative marsh elevation ($Z^*$) in the summer, regardless of tidal stage, and the fall, but only during periods of exposure (Fig. 2a, c, f; Table 1B). In contrast, temperatures at 50 cm were less sensitive to marsh elevation within the tidal frame.



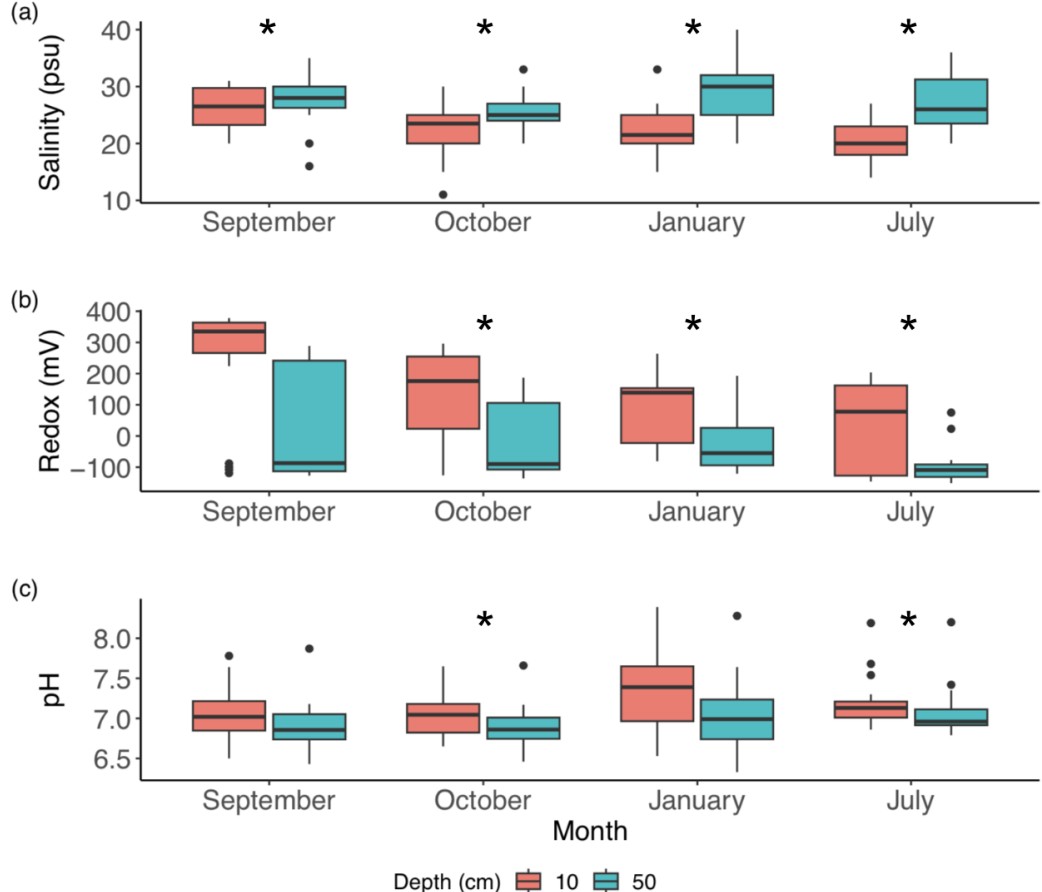

Figure 1. Soil porewater salinity (a), redox (b), and pH (c) pooled across transects at 2, 3, 6, and 12 months (September, October, January, and July, respectively) at 10 cm (red) and 50 cm (blue) depth. Significant differences (p<0.05) between depths are denoted with an asterisk (*).

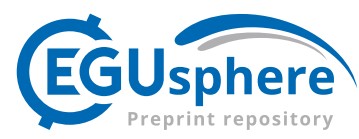

Table 1. Soil temperatures by depth and season and with relative position in the tidal frame (Z*). (a). Average (± SE) temperatures (°C) by depth, season, and during periods of tidal inundation or exposure. Significant differences between tidal stages within a season are denoted by * (p<0.05). (b). Linear regression models tested temperature changes with relative marsh surface elevation (Z*) over tidal stages and seasons. Correlations between temperature and relative elevation (Z*) are described by the slope coefficients, associated p values, and the multiple (mult) and adjusted (adj) $r^2$, that reflect the variance explained by fixed effects alone or by fixed and random effects combined, respectively. Significant differences (<0.05) between tidal stages within a depth and season are denoted by superscripts.

**(a)**

| | 10 cm | | | | | | 50 cm | | | | | |
|---|---|---|---|---|---|---|---|---|---|---|---|---|
| | *Summer* | | *Fall* ** | | *Winter* * | | *Summer* | | *Fall* ** | | *Winter* * | |
| | Exposed | Inundated | Exposed | Inundated | Exposed | Inundated | Exposed | Inundated | Exposed | Inundated | Exposed | Inundated |
| Temp. (°C) | 28.19 ± 0.08 | 28.21 ± 0.06 | 19.58 ± 0.09 | 20.95 ± 0.05 | 15.65 ± 0.06 | 15.83 ± 0.04 | 27.25 ± 0.09 | 27.21 ± 0.09 | 21.6 ± 0.05 | 22.57 ± 0.05 | 16.83 ± 0.04 | 16.68 ± 0.05 |

**(b)**

| Depth (cm) | Season | Flooding | Relative Elevation (Z*) | | R² | |
|---|---|---|---|---|---|---|
| | | | Slope | p-value | Mult. R² | Adj. R² |
| 10 cm | Summer | Exposed | -0.020 ± 0.006 [a] | **0.01** | 0.45 | 0.41 |
| | | Inundated | -0.017 ± 0.004 [a] | **0.00** | 0.60 | 0.56 |
| | Fall | Exposed | -0.029 ± 0.011 [a] | **0.02** | 0.37 | 0.32 |
| | | Inundated | 0.013 ± 0.005 [b] | **0.02** | 0.36 | 0.30 |
| | Winter | Exposed | 0.005 ± 0.011 [a] | 0.65 | 0.02 | -0.06 |
| | | Inundated | -0.011 ± 0.007 [a] | 0.15 | 0.16 | 0.09 |
| 50 cm | Summer | Exposed | -0.012 ± 0.008 [a] | 0.16 | 0.14 | 0.08 |
| | | Inundated | -0.014 ± 0.007 [a] | 0.09 | 0.21 | 0.15 |
| | Fall | Exposed | -0.009 ± 0.006 [a] | 0.16 | 0.15 | 0.08 |
| | | Inundated | 0.013 ± 0.004 [a] | **0.01** | 0.45 | 0.40 |
| | Winter | Exposed | 0.007 ± 0.007 [a] | 0.30 | 0.08 | 0.01 |
| | | Inundated | -0.008 ± 0.008 [b] | 0.36 | 0.06 | -0.01 |

$\log_{10}$ Temp. (°C)





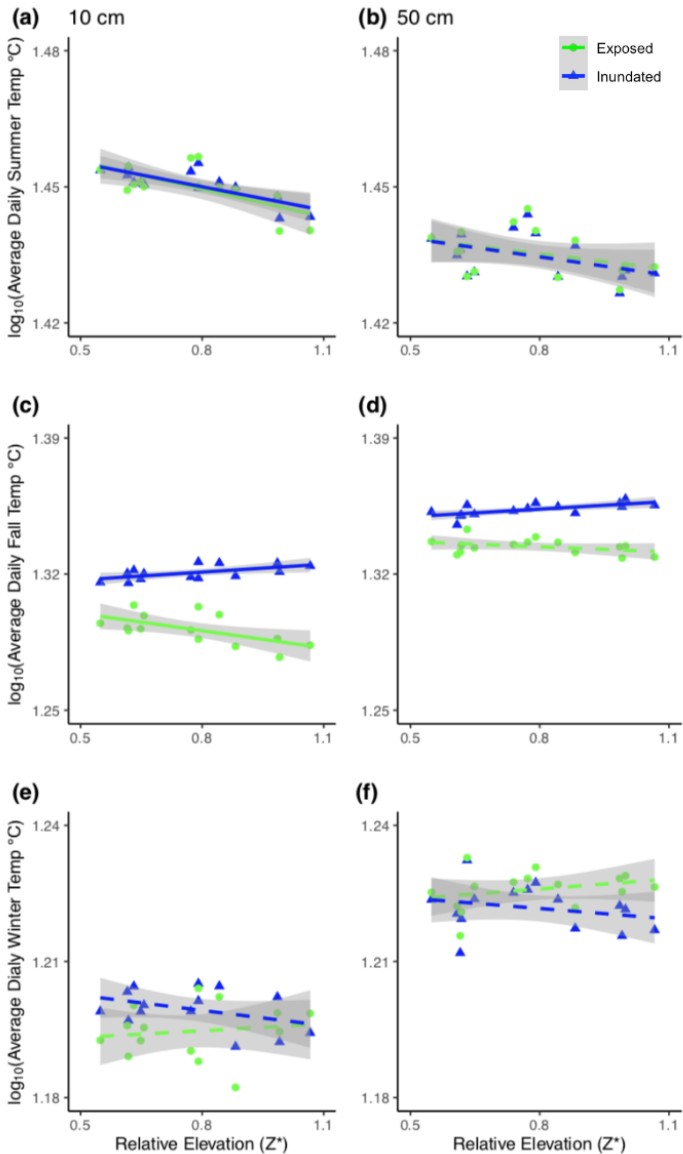

Figure 2. Soil temperatures at 10 cm (left) and 50 cm (right) changed with tidal stage (inundated, blue; exposed, green) and relative mash surface elevation in the tidal frame (Z*) during summer (top), fall (middle), and winter (bottom). Correlations with $p < 0.05$ or $> 0.05$ are denoted with solid or dashed lines, respectively. See table 1 for statistical results.





### 3.2 Decay Rates

TBI decay rates ($k$, $d^{-1}$) decreased with depth and time (Fig. 3 a, b; Table 3). Rates were 50%, 32%, and 118% faster in the 10 cm horizon at the 3-, 6-, and 12- month time points, respectively, compared to 50 cm depth. Decay rates slowed to a similar degree between 3- and 6-months at 10 cm (60%) and 50 cm (54%) but there was less of a slowdown between 6- and 12-months at 10 cm (17%) compared to 50 cm (50%; Table 3A). This translated into a 3-fold slowing

of turnover times from 140 to 416 days at 10 cm but a 4-fold slowing from 209 to 903 days at 50 cm over the year-long experiment (Table 3A). Slowing decay rates between 10 and 50 cm and over time were unlikely due to the stabilization factor (S) which was similar at both depths and decreased from 6 to 12 months (Table 3A).

   Tea-specific empirical decay rates ($k_g$, $k_r$), bookended TBI $k$, varied significantly in

comparison to TBI decay rates. Green tea rates were 67-162% and 150-327% higher and rooibos tea rates were 48-64% and 34-63% lower at 10 cm and 50 cm, respectively than TBI rates (Fig. 3c-f; Table 3). The percent difference between shallow and deeper rates was much greater for rooibos (29-36%) than green tea (0-2.7%), but both tea types slowed to similar extents at 10 cm and 50 cm between 3-6 months (37-38% green, 39-42% rooibos) and 6-12 months (37-38% green,

34-35% rooibos).

   We next compared decomposable fractions (a) and stabilization factors (S) for green and rooibos teas. Variables estimated with the TBI and empirical approaches were the same for green tea, but not for rooibos (equations 2-5). The TBI $a_r$ values, calculated from $H_g$ and $S_g$ (equation 4), were 71-200% higher than empirical $a_r$ values, based on the mass fraction of rooibos tea lost at

each time point and depth (Table 3). Stabilization factors for rooibos tea ($S_r$) were 247-285% and 279-423% greater than for green tea (S, $S_g$) at 10 cm and 50 cm, respectively (Table 3). For both tea types S values were lowest at 12 months, but only $S_r$ had lower values at the deeper depth. This demonstrates that rooibos tea was more sensitive than green tea to differences in the soil environment between 10 and 50 cm over the time scales in this experiment.

Root decay rates estimated from litterbags buried in a nearby *S. alterniflora* marsh ranged from 0.0015-0.0021 $d^{-1}$, and were slightly faster in the interior marsh plain compared to the creekbank levee (Table S2). These rates are slower than decay estimates calculated from the TBI approach and green tea ($k_g$) but are comparable to rooibos tea ($k_r$) loss rates (Table 3).



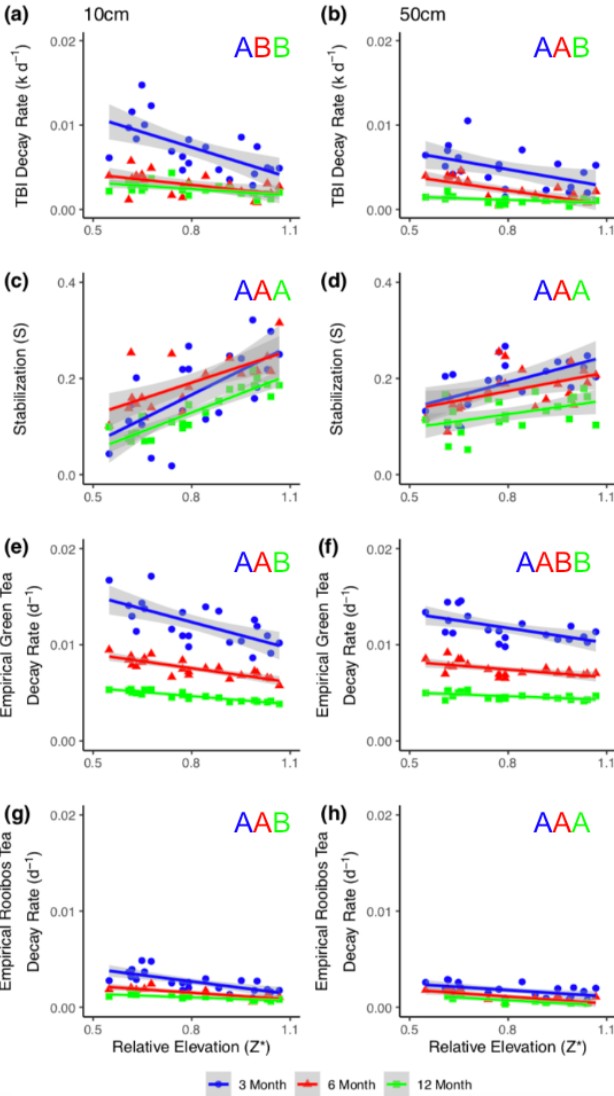

Figure 3. TBI decay rates (a, b) and stabilization factors (c, d) and empirical decay rates (e, f, g, h) at 10 cm (left) and 50 cm right). Decay rates decreased while stabilization factors increased with relative marsh surface elevation within the tidal frame (Z*) at both depths. Significant correlations ($p<0.05$) are denoted with solid lines. Contrasts between k and S at the 3- (blue), 6- (red), and 12- (green) time points are denoted by letters of the same color. See Table 2 for statistical results.





Table 2. Tea bag index (TBI) and empirical decay rates across soil depths and deployment times. Linear mixed effect models were used to test decay and stabilization responses to the depth (10 or 50 cm) and length (3, 6, or 12 months) of deployments. The marginal (mar.) and conditional (cond.) $r^2$ reflect the variance explained by fixed effects alone or by fixed and random effects combined, respectively. Significant p values (< 0.05) are in bold.

| Response | Depth | | Time | | Depth x Time | | $R^2$ | |
|---|---|---|---|---|---|---|---|---|
| | *Slope* | *p* | *Slope* | *p* | *Slope* | *p* | *Mar.* | *Cond.* |
| TBI k ($d^{-1}$) | $-7.1 \times 10^{-5} \pm 1 \times 10^{-5}$ | **<0.001** | $-5.0 \times 10^{-4} \pm 7 \times 10^{-5}$ | **<0.001** | $5.3 \times 10^{-6} \pm 2 \times 10^{-6}$ | **0.01** | 0.35 | 0.54 |
| Empirical $\log_{10}$(Green $k_g$ ($d^{-1}$)) | $-4.5 \times 10^{-5} \pm 4 \times 10^{-4}$ | **0.002** | $-4.3 \times 10^{-2} \pm 2 \times 10^{-3}$ | **<0.001** | $-1.3 \times 10^{-5} \pm 5 \times 10^{-5}$ | 0.80 | 0.87 | 0.92 |
| Empirical Rooibos $k_r$ ($d^{-1}$) | $-2.2 \times 10^{-5} \pm 5 \times 10^{-6}$ | **<0.001** | $-1.7 \times 10^{-4} \pm 2 \times 10^{-5}$ | **<0.001** | $1.1 \times 10^{-6} \pm 7 \times 10^{-7}$ | 0.10 | 0.39 | 0.74 |
| TBI S (same as $S_g$) | $4.4 \times 10^{-4} \pm 4 \times 10^{-4}$ | 0.95 | $-4.6 \times 10^{-3} \pm 2 \times 10^{-3}$ | **<0.001** | $-6.0 \times 10^{-5} \pm 5 \times 10^{-5}$ | 0.24 | 0.17 | 0.47 |
| Rooibos tea stabilization ($S_r$) | $1.8 \times 10^{-3} \pm 9 \times 10^{-4}$ | **<0.001** | $-1.4 \times 10^{-2} \pm 4 \times 10^{-3}$ | **<0.001** | $1.7 \times 10^{-4} \pm 1 \times 10^{-4}$ | 0.12 | 0.21 | 0.63 |



Table 3. Average ± SE of decomposable fractions (a) and stabilization factors used to estimate decay rates (k) and turnover times (days) with the TBI (a) and empirical (b) approaches. Significant differences (p>0.05) between deployment times within a depth horizon are denoted by superscripts.

**(a) TBI Calculations**

| Depth (cm) | Time (months) | $a_r$ | S (or $S_g$) | $k$ (d$^{-1}$) | Turnover time (d) |
|---|---|---|---|---|---|
| 10 | 3 | $0.46 \pm 0.02$ | $0.17 \pm 0.02$ [a] | $7.2 \times 10^{-3} \pm 7 \times 10^{-4}$ [a] | $140 \pm 15$ |
| | 6 | $0.45 \pm 0.01$ | $0.19 \pm 0.01$ [a] | $2.9 \times 10^{-3} \pm 3 \times 10^{-4}$ [b] | $341 \pm 66$ |
| | 12 | $0.48 \pm 0.01$ | $0.13 \pm 0.01$ [b] | $2.4 \times 10^{-3} \pm 2 \times 10^{-4}$ [b] | $416 \pm 33$ |
| 50 | 3 | $0.45 \pm 0.01$ | $0.19 \pm 0.01$ [a] | $4.8 \times 10^{-3} \pm 5 \times 10^{-4}$ [a] | $209 \pm 33$ |
| | 6 | $0.45 \pm 0.01$ | $0.18 \pm 0.01$ [a] | $2.2 \times 10^{-3} \pm 3 \times 10^{-4}$ [b] | $464 \pm 105$ |
| | 12 | $0.48 \pm 0.01$ | $0.13 \pm 0.01$ [b] | $1.1 \times 10^{-3} \pm 9 \times 10^{-5}$ [c] | $903 \pm 115$ |

**(b) Empirical Calculations**

| Tea Type | Depth (cm) | Time (months) | $a_r$ or $a_g$ | $S_r$ | empirical $k_g$, $k_r$ (d$^{-1}$) | Turnover time (d) |
|---|---|---|---|---|---|---|
| Green | 10 | 3 | $0.70 \pm 0.02$ [a] | | $1.2 \times 10^{-2} \pm 5 \times 10^{-4}$ [a] | $85 \pm 3$ |
| | | 6 | $0.68 \pm 0.01$ [a] | | $7.6 \times 10^{-3} \pm 2 \times 10^{-4}$ [b] | $134 \pm 4$ |
| | | 12 | $0.73 \pm 0.01$ [b] | | $4.7 \times 10^{-3} \pm 1 \times 10^{-4}$ [c] | $217 \pm 5$ |
| | 50 | 3 | $0.68 \pm 0.01$ [a] | | $1.2 \times 10^{-2} \pm 3 \times 10^{-4}$ [a] | $86 \pm 2$ |
| | | 6 | $0.69 \pm 0.01$ [a] | | $7.4 \times 10^{-3} \pm 2 \times 10^{-4}$ [b] | $136 \pm 3$ |
| | | 12 | $0.74 \pm 0.01$ [b] | | $4.7 \times 10^{-3} \pm 7 \times 10^{-5}$ [c] | $216 \pm 4$ |
| Rooibos | 10 | 3 | $0.22 \pm 0.02$ [a] | 0.59 | $2.6 \times 10^{-3} \pm 2 \times 10^{-4}$ [a] | $440 \pm 40$ |
| | | 6 | $0.18 \pm 0.02$ [b] | 0.68 | $1.5 \times 10^{-3} \pm 1 \times 10^{-4}$ [b] | $785 \pm 103$ |
| | | 12 | $0.28 \pm 0.01$ [c] | 0.50 | $9.8 \times 10^{-4} \pm 6 \times 10^{-5}$ [c] | $1083 \pm 75$ |
| | 50 | 3 | $0.16 \pm 0.01$ [a] | 0.72 | $1.8 \times 10^{-3} \pm 1 \times 10^{-4}$ [a] | $697 \pm 78$ |
| | | 6 | $0.15 \pm 0.02$ [a] | 0.74 | $1.1 \times 10^{-3} \pm 1 \times 10^{-4}$ [b] | $1168 \pm 165$ |
| | | 12 | $0.18 \pm 0.02$ [a] | 0.68 | $7.3 \times 10^{-4} \pm 1 \times 10^{-4}$ [c] | $1703 \pm 248$ |



### 3.3 Biotic and abiotic variables correlated with TBI decay rates

TBI decay rates at 10 and 50 cm depth negatively correlated with relative elevation ($Z^*$) across all time points (Table 4). Rates also correlated with other abiotic and biotic factors but, unlike $Z^*$, none of these relationships were consistently significant at both depths and throughout the experiment. For instance, porewater salinity correlated with TBI decay at 50 cm, but not 10 cm. Variables reflecting certain plant (stem height, above- and below-ground biomass) and soil (stiffness) characteristics consistently correlated with decay rates at 3- and 6- months but not at 12- months at both depths. Other variables representing bioturbation (crab burrows), grazing (snails) and porewater chemistry (pH, redox) correlated sporadically, if at all, with TBI decay. The directionality of many of these correlations (but not statistical significance) remained unchanged over the 12-month period and we suspect this is likely because, like decay, they responded to relative marsh elevation ($Z^*$). To explore this further, we evaluated correlations between relative elevation ($Z^*$) and aboveground plant biomass and soil stiffness. Negative relationships with aboveground biomass ($r^2 = 0.60$, $p<0.05$) point to lower grass production at higher elevations, where there is less tidal flooding. Positive correlations with soil stiffness ($r^2 = 0.21$, $p<0.05$) are consistent with less consolidation (i.e., greater porewater flushing) or other gradients in soil properties (e.g., grain size, which affects burrowing) at lower elevations with greater flooding. Because plant and soil properties affect decay (Liu et al., 2008; Noyce et al., 2023), and we cannot separate those drivers from relative marsh surface elevation ($Z^*$), we focus on how decay changes with position in the tidal frame from here forward.



Table 4. Spearman rank correlations between TBI decay rates ($k$, $d^{-1}$) and potential abiotic and biotic drivers at 10 and 50 cm depth and for the three deployment intervals (3, 6, or 12 months). Significant coefficients ($p < 0.05$) are bolded.

| Response | 10 cm | | | 50 cm | | |
|---|---|---|---|---|---|---|
| | 3 mon. | 6 mon. | 12 mon. | 3 mon. | 6 mon. | 12 mon. |
| Relative marsh surface elevation (Z*) | **-0.60** | **-0.52** | **-0.64** | **-0.49** | **-0.58** | **-0.44** |
| Crab burrows (count m$^{-2}$) | -0.27 | **-0.59** | -0.16 | -0.13 | **-0.43** | -0.32 |
| Snails (ind m$^{-2}$) | -0.30 | -0.16 | 0.33 | -0.41 | -0.37 | -0.23 |
| Spartina stem density (shoots m$^{-2}$) | -0.39 | -0.37 | 0.18 | -0.4 | **-0.53** | **-0.41** |
| Spartina stem height (cm) | **0.57** | **0.53** | 0.09 | **0.62** | **0.80** | **0.60** |
| Spartina aboveground biomass (g m$^{-2}$) | **0.54** | **0.61** | **0.60** | **0.46** | **0.66** | 0.34 |
| Spartina root biomass (g cm$^{-3}$) | 0.38 | 0.22 | -0.07 | 0.22 | **0.45** | 0.12 |
| Spartina rhizome biomass (g cm$^{-3}$) | **0.60** | **0.45** | **0.54** | **0.61** | **0.58** | 0.29 |
| Soil stiffness (kpa) | **-0.60** | **-0.71** | -0.33 | **-0.53** | **-0.78** | **-0.63** |
| Porewater salinity (PSU) | -0.41 | -0.45 | -0.15 | **-0.43** | **-0.79** | **-0.60** |
| Porewater pH | 0.25 | **0.57** | 0.42 | 0.42 | 0.28 | 0.29 |
| Porewater redox (Eh) | -0.33 | 0.15 | -0.07 | 0.26 | 0.32 | 0.24 |



### 3.4 Decay, Stabilization, and Relative Marsh Surface Elevation (i.e., Z*)

TBI and empirical decay rates decreased while TBI S increased at higher relative elevations
(Z*; Table 5, Fig.3). Changes in TBI and empirical green tea decay rates with relative elevation
(Z*) at 10 cm and 50 cm were sharpest at three months and became less pronounced with time.
Further, these gradients were generally steeper at 10 cm and more gradual at 50 cm. In contrast,
correlations between relative elevation (Z*) and empirical rooibos decay rates were more similar
between soil depths and stable over time. Correlations with relative elevation (Z*) generally
accounted for greater fractions of the variability in empirical green ($r^2$ = 0.44-0.86) and rooibos ($r^2$
= 0.50-0.75) rates compared to TBI decay ($r^2$ = 0.33-0.41) at 10 cm. The explanatory power of
relative elevation (Z*) was lower at 50 cm for the empirical rates but differences between depths
were less clear for TBI rates. These results demonstrate that flooding effects, approximated by
relative marsh surface elevation (Z*), on decay rates are stronger at 10 cm, but still apparent at
deeper horizons, and persist for up to one year.

The TBI S factors had the opposite relationship with relative elevation (Z*) and increased
at higher elevations but these correlations did not change over time at either 10 or 50 cm depth
(Fig. 3c-d; Table 4). The change in TBI S with relative elevation (Z*) was greater at 10 cm
compared to 50 cm and these correlations were largely constant throughout the experiment (Table
5). A greater fraction of variability in TBI S (i.e., $r^2$) could be attributed to relative elevation (Z*)
at 10 cm, suggesting that the factors affecting S and related to tidal inundation are strongest in the
rhizosphere.



Table 5. Decay rates ($k$, $d^{-1}$) decreased and stabilization factors (S) increased relative to marsh surface position in the tidal frame (Z*) at both depths and throughout the experiment. Differences between slopes across the three time points within a depth horizon are denoted by superscripts.

| Response | Depth (cm) | Time (months) | Relative Elevation (Z*) | | $r^2$ |
|---|---|---|---|---|---|
| | | | *Slope* | *p* | |
| TBI Decay rate ($k$ $d^{-1}$) | 10 | 3 | $-1.2 \times 10^{-2} \pm 3 \times 10^{-3}$ [a] | <0.01 | 0.41 |
| | | 6 | $-4.1 \times 10^{-3} \pm 2 \times 10^{-3}$ [b] | 0.02 | 0.27 |
| | | 12 | $-2.6 \times 10^{-3} \pm 8 \times 10^{-4}$ [b] | <0.01 | 0.33 |
| | 50 | 3 | $-6.8 \times 10^{-3} \pm 3 \times 10^{-3}$ [a] | 0.02 | 0.25 |
| | | 6 | $-5.6 \times 10^{-3} \pm 1 \times 10^{-3}$ [a] | <0.01 | 0.47 |
| | | 12 | $-1.2 \times 10^{-3} \pm 5 \times 10^{-4}$ [b] | 0.03 | 0.24 |
| TBI Stabilization | 10 | 3 | $0.34 \pm 0.09$ [a] | <0.01 | 0.43 |
| | | 6 | $0.22 \pm 0.06$ [a] | <0.01 | 0.48 |
| | | 12 | $0.26 \pm 0.03$ [a] | <0.01 | 0.82 |
| | 50 | 3 | $0.18 \pm 0.06$ [a] | <0.01 | 0.35 |
| | | 6 | $0.13 \pm 0.05$ [a] | 0.02 | 0.25 |
| | | 12 | $0.10 \pm 0.04$ [a] | 0.03 | 0.21 |
| Empirical Green tea decay ($k_g$ $d^{-1}$) | 10 | 3 | $-9.2 \times 10^{-3} \pm 2 \times 10^{-3}$ [a] | <0.01 | 0.44 |
| | | 6 | $-4.9 \times 10^{-3} \pm 7 \times 10^{-4}$ [a] | <0.01 | 0.68 |
| | | 12 | $-2.8 \times 10^{-3} \pm 3 \times 10^{-4}$ [b] | <0.01 | 0.86 |
| | 50 | 3 | $-5.2 \times 10^{-3} \pm 2 \times 10^{-3}$ [a] | <0.01 | 0.35 |
| | | 6 | $-2.7 \times 10^{-3} \pm 8 \times 10^{-4}$ [ab] | <0.01 | 0.38 |
| | | 12 | $-1.3 \times 10^{-3} \pm 4 \times 10^{-4}$ [b] | <0.01 | 0.34 |
| Empirical Rooibos tea decay ($k_r$ $d^{-1}$) | 10 | 3 | $-4.4 \times 10^{-3} \pm 1 \times 10^{-3}$ [a] | <0.01 | 0.50 |
| | | 6 | $-2.4 \times 10^{-3} \pm 5 \times 10^{-4}$ [a] | <0.01 | 0.67 |
| | | 12 | $-1.2 \times 10^{-3} \pm 2 \times 10^{-4}$ [b] | <0.01 | 0.75 |
| | 50 | 3 | $-2.2 \times 10^{-3} \pm 8 \times 10^{-4}$ [a] | 0.01 | 0.27 |
| | | 6 | $-2.4 \times 10^{-3} \pm 6 \times 10^{-4}$ [a] | <0.01 | 0.55 |
| | | 12 | $-1.9 \times 10^{-3} \pm 5 \times 10^{-4}$ [a] | <0.01 | 0.66 |



## 4. Discussion

### 4.1 Methodological considerations.

Decay rates based on the TBI and empirical green and rooibos mass losses slowed over time and with depth and were fastest in plots sitting lower in the tidal frame (Fig. 3, Table 3, 5). Average green tea decay rates were 67-327% faster than TBI k which was 34-64% faster than rooibos rates (Table 3). Between soil depths, TBI and rooibos decay rates were 30-118% and 34-44% faster, respectively, at 10 cm compared to 50 cm whereas empirical green tea rates were nearly equivalent (0-3% change; Table 3). Differences between TBI and empirical decay rates at both depths and, for rooibos tea, with time were relatively constant while the gap between TBI and green tea widened over the year. These patterns suggest that, at the plot scale and over one year, TBI rates reflect rooibos tea dynamics (oxidized needles and branches from the rooibos bush) slightly more than green tea (*Camellia sinensis* leaves and buds).

At a larger scale across marsh surface elevations, changes in TBI k with relative elevation (Z*) closely mirrored empirical green tea dynamics, especially in the shallower soil horizon (Table 5; Fig. 3). Steeper drops in TBI decay with relative elevation (Z*) during the first three months were driven by changes in green tea mass loss and likely influenced by leaching due to greater tidal flushing of porewater at lower elevations (Fig. 3, Table 5). In contrast, mass loss rates of rooibos tea changed less across relative marsh surface elevations (Z*) and were more constant over time. These patterns are consistent with a short-term leaching experiment demonstrating faster losses of green tea (10-50%) than rooibos (<5 – 20%) and greater sensitivity to temperature, water turnover, and soil moisture content (Lind et al., 2022). We cannot isolate the magnitude of leaching effects from microbial decomposition since both would have occurred during the first several months. That leaching accelerates decay is not a problem exclusive to the TBI - it also affects interpretation of mass loss rates from litterbags with local detritus (Cotrufo et al., 2010; Gessner et al., 1999; MacDonald et al., 2018; Seelen et al., 2019) - but the potential magnitude of abiotic loss highlights that decay coefficients from 3-month deployments, as prescribed by Keuskamp et al. (2013), should not be interpreted solely as a function of the microbial community. Instead, extending the duration and increasing the number of sampling points of the TBI could result in decay rates that are more representative of microbial processing (Lind et al., 2022; Marley et al., 2019).

Another argument for extending the duration of TBI studies in marshes is that mass loss rates of green tea did not plateau after three months. Green tea mass loss increased by 4.3% at 10 cm and 8.8% at 50 cm over the final nine months of the experiment (Table 3b, $a_g$ values). The fraction of green tea mass loss was never greater than $H_g$ (hydrolysable fraction) reported by Keuskamp et al. (2013), as has happened in short-term leaching studies (Lind et al., 2022) and forest soils (Mori et al., 2022). As a result, TBI S did not fall below zero, which would have skewed the TBI $a_r$. Choice of $H_r$ and $H_g$ values is important because the





hydrolysable fraction, operationally defined as the sum of nonpolar, water soluble, and acid soluble compounds, is sensitive to methodology (e.g., Mueller et al., 2018) and affects S.

A central tenant of the TBI is that S asymptotes at three months and values are the same for green and rooibos teas (Keuskamp et al., 2013). However, we found that TBI S (equivalent to $S_g$) and $S_r$ decreased from 3-to-12 months and values were 247-423% higher for rooibos than green tea (Table 3a-b). Moreover, differences between $S_g$ and $S_r$ increased over time. The caveat that $S_g$ and $S_r$ are not equal is not a function of the marsh environment as Mori et al. (2022) also report differences across four temperate forest stands.

The assertion that S is the same for green and rooibos teas rests on the assumption that stabilization is controlled by environmental factors (Keuskamp et al., 2013) and independent of compositional differences that affect organic matter-soil interactions (e.g., mineral association, incorporation into aggregates, etc.) and, as a result, decomposition rates (Marschner et al., 2008; Mikutta et al., 2006). Yet, easily degradable non-structural compounds can be preserved over long time scales due to physio-chemical interactions while

complex macromolecules are not intrinsically recalcitrant (Dungait et al., 2012; Kallenbach et al., 2016; Mikutta et al., 2006). As such, there appears to be limited theoretical support for S as formulated by the TBI.

Comparisons between TBI and empirical decay rates averaged across plots demonstrate that this index is weighted slightly more by rooibos mass loss rather than being an equal blend of both teas. However, when distributed across the environmental gradient of tidal flooding (i.e., Z*) the different sensitivities of

each tea type became more apparent. Faster mass loss rates of green and rooibos teas, which are rich in soluble tannins and aromatic compounds associated with lignin monomers, respectively (Duddigan et al., 2020), in lower relative elevation (Z*) plots are consistent with leaching in the short-term (three months) and the effects of porewater turnover on decomposition in the longer term (Fig. 3). Running experiments beyond three months and increasing sampling intervals will likely allow for better distinctions between leaching and

decomposition effects (Duddigan et al., 2020; Lind et al., 2022; Marley et al., 2019). The TBI was developed for terrestrial soils and our results demonstrate the some assumptions need to be carefully assessed when applying this method to saturated, wetland soils. Knowing the different sensitivities of green and rooibos teas to physical, chemical, and biological processes is valuable for interpreting controls on organic matter mass loss rates across environmental gradients and different ecosystems.


### 4.2 Decay rate context

Organic matter decay rates estimated by the TBI were higher than previous measurements of 0.0010 – 0.0026 d$^{-1}$ from Georgia's minerogenic salt marshes based on field litterbags and laboratory leaching and incubation experiments conducted over 150 – 540 days (Benner et al., 1984; Benner et al. 1987; Benner et

al., 1991; Rice & Tenore, 1981) (Table S2). The slowest rates were based on lignin while faster rates were estimated from losses of structural polysaccharides (cellulose, hemicellulose) or plant tissue mass. The



highest decay rate was calculated from polysaccharides in root and rhizome litter (Benner et al., 1991) and was 30-73% faster than root mass loss along creekbank levees (0.0015 d$^{-1}$) and marsh interiors (0.0020 d$^{-1}$; Table S2). The TBI rates at three months are 2.8 – 7.2 times faster than prior studies but that drops to roughly

double over longer, 6-12 month periods (10 cm horizon only; Table 3; Fig 3), with the exception of the rapid polysaccharide-specific rate (Benner et al., 1991). This is perhaps not surprising since nuclear magnetic resonance (NMR) spectroscopy demonstrates sharp reductions in O-alkyl compounds consistent with carbohydrates and polysaccharides and aromatic compounds consistent with tannins during green tea incubations (Duddigan et al., 2020). Higher TBI rates could also reflect differences in the preparation and

processing (e.g., milling, oxidation) of the organic matter filling tea and litter bags. Although the TBI overestimates decay, empirical rooibos mass loss rates are more consistent with natural marsh litter (Table 4; S2). Regardless, these comparisons suggest that rooibos tea may adequately mimic decay dynamics of local litter, depending on study goals, but combining with green tea in the TBI results in accelerated rates.

We expected TBI rates to be faster in Georgia compared to higher latitudes based on the metabolic

theory of ecology, which predicts that decay rates increase with rising temperatures (Yvon-Durocher et al., 2010), and observations that warming accelerates loss of labile compounds in soils (Conant et al., 2011; Melillo et al., 2002). To test this, we compared our rates with those from 7 other salt marsh TBI studies that encompass 11 countries and span a latitudinal gradient of 93.7° (-37.7°-56°); Fig. 4, SI table 2) (Mueller et al., 2018, Puppin et al., 2023, Marley et al., 2019, Alsafran et al., 2017, Yousefi Lalimi et al., 2018,

Sanderman and Eagle, unpub, Tang et al., 2023). North America accounted for 50% of the observations, and only one observation came from the southern hemisphere. Teabags were buried at 8 cm in most of those studies, per Keuskamp et al. (2013), whereas we used a 10 cm depth. It is unlikely that this slight difference in burial depth skews comparisons since both are within the rooting zone. Marley et al. (2019) used locally sourced tea, rather than the prescribed Lipton brand, but reported that the two are compositionally similar.

Potentially more important is that most studies used H$_g$ and H$_r$ values reported by Keuskamp et al. (2013) while Tang et al. (2023) performed different extractions to derive their own estimates of hydrolysable fractions and didn't provide estimations of decay based on the original TBI constants. Differences in H values across studies are relatively minor and would more strongly affect S (stabilization) than k (decay). Our Sapelo Island, GA, 3-month rates were similar to other temperate salt marshes in North Carolina, Virgina, Maryland,

California, and Massachusetts, USA and Zeijhong Province (ZJ), China (Fig. 4). The lack of a directional trend within these latitudes contrasts with small-to-moderate warming effects on marsh litter decomposition in field experiments (Charles & Dukes, 2009; Tang et al., 2023) and across spatial gradients (Kirwan et al., 2014). Sapelo Island, GA, rates inconsistently related to those measured at higher and lower latitudes (Fig. 4). The absence of a general latitudinal trend (p>0.05, r$^2$=0.01) for 3-month decay rates could reflect

interactions within the soil environment that affect decomposition, such as leaching, tidal flushing, redox





conditions, salinity, mineral associations that protect organic matter and alter its kinetic properties (Conant et al., 2011; Craine et al., 2010), and plant root exudates, among other variables (Fettrow et al., 2023; Keiluweit et al., 2015; Seyfferth et al., 2020; Spivak et al., 2023). It is also possible that 3-month rates are more sensitive to leaching than temperature in saturated marsh soils as differences between sampling time points (3-, 6-, or 12- months) are often as great as between latitudes (e.g., Sapelo Island, GA; Schleswig-Holstein (SH), Germany; East Lothian (ELN), Scotland). It is possible that latitudinal trends may become more apparent following longer teabag deployments when microbial processing would be the dominant control on organic matter loss.

The TBI's stabilization factor (S) is meant to represent the process by which labile compounds become refractory under certain environmental conditions (Keuskamp et al., 2013) and should increase over time as decay progresses (Marschner et al., 2008; Mikutta et al., 2006). A slight, negative correlation between 3-month TBI S and k values ($p<0.05$, $r^2 =0.11$) across the compiled data from all seven studies supports this prediction (Fig. 4). However, the negative relationship between TBI S and k is not universal across individual studies (Keuskamp et al., 2013; Seelen et al., 2019). On Sapelo Island (GA, USA), the highest S values coincided with the fastest decay rates in the first 3 months (Fig. 4). Stabilization values then decreased between 3- and 6- months but there was no overall temporal trend because values increased at 12-months. This variability is not unique to our site; S values decreased and increased at East Lothian (ELN), Scotland and Schleswig-Holstein (SH), Germany, respectively, between 3- and 12- months. The absence of a clear latitudinal gradient in TBI S values suggests that this proxy is largely insensitive to global-scale temperature gradients (Fig. 4). Yet, this contrasts with Mueller et al. (2018), who reported higher S values at higher latitudes along the North American Atlantic coast. The mixed relationships between TBI S with k and across latitudes could reflect variability in the many physicochemical processes that affect microbial access to organic matter. This interpretation, though, is caveated by the finding here and discussed further by Mori et al. (2022) that green and rooibos teas do not share S values which violates assumptions of the TBI method (Table 3).



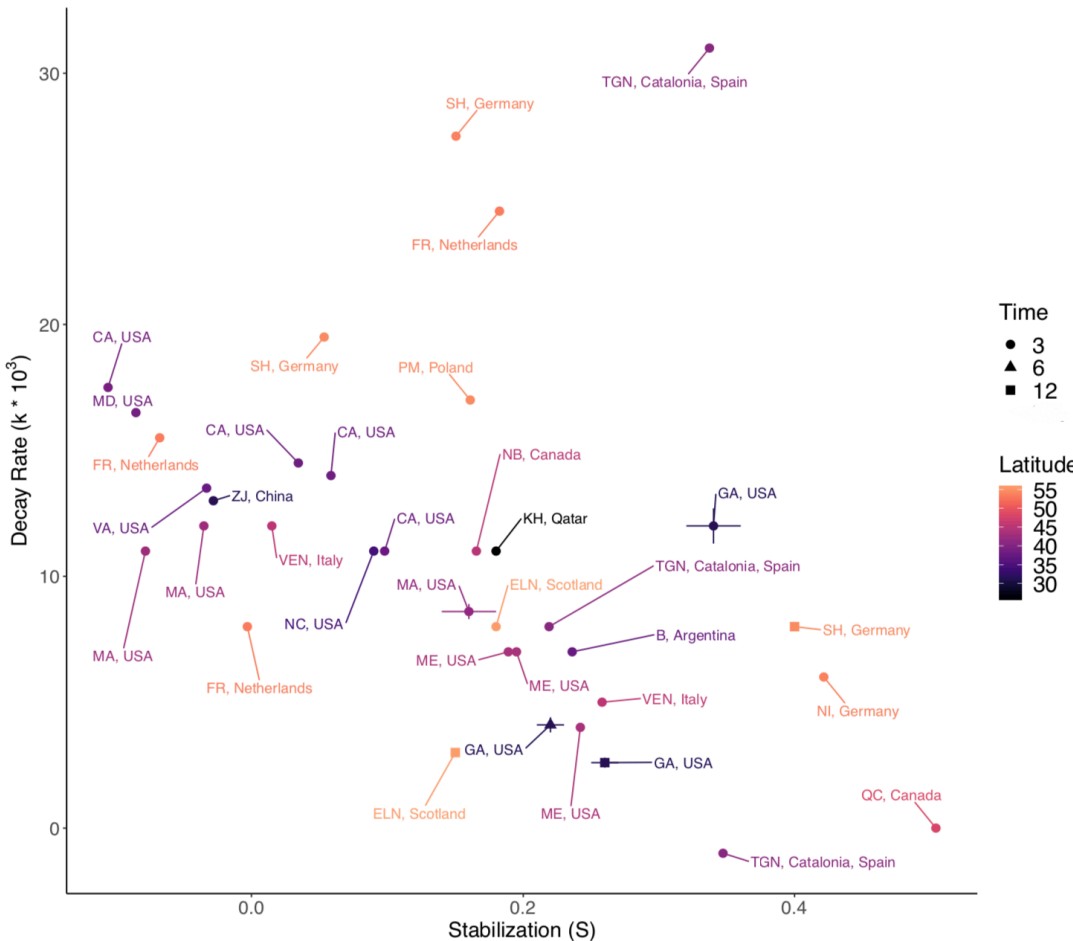

Fig. 4. Average TBI decay rates and stabilization factors in salt marshes as reported by seven published and
unpublished data sets spanning a latitudinal gradient of 56.0° N to 37.7° S (see SI Table 2). To facilitate
comparison across northern and southern latitudes all coordinates are represented as positive values. Data are
from 3-, 6-, or 12-months of burial at 8-10 cm depth. The GA, USA data are from this study.

Faster decay rates estimated using the TBI method relative to more conventional litterbag and
laboratory experiments suggest that these approaches are not interchangeable. It is possible that tea
processing, including oxidation and milling into small pieces, increases vulnerability to decomposition and
that microbes respond strongly to compositional differences between allochthonous organic matter and local
marsh detritus. Similar decay rates between rooibos tea and more conventional approaches suggest that this
aspect of the TBI could be a reasonable proxy when the experimental goal is to assess drivers independently
of site-specific differences in organic matter composition and material preparation. Few studies have directly



compared decay rates from the TBI, its components, and more conventional approaches but this would be useful in assessing whether Keuskamp et al.'s (2013) method can be applied broadly, in dry and saturated soils. If rooibos tea decay rates from 6-month and longer deployments are comparable to more conventional approaches, then clearer patterns between marshes and across latitudes may become more apparent.

### 4.3 Decay within and below the rhizosphere.

The trajectory of rapid initial TBI decay rates followed by progressive slowing is consistent with decomposition models (Morris & Bowden, 1986; Valiela et al., 1985), litter bags (Benner et al., 1991) and depth profiles of marsh soil organic matter (Luk et al., 2021) and likely reflects initial losses of soluble and bioavailable compounds and relative accumulation of larger macromolecules (Benner et al., 1991; Marley et al., 2019; Moran et al., 1989; J. O. Wilson et al., 1986). Slower rates at deeper depths are consistent with a

more stable environment relative to the rhizosphere where root oxygen loss and exudates, bioturbation, and porewater flushing are associated with faster decay (Furukawa et al., 2004; Li et al., 2021; C. A. Wilson et al., 2012). However, negative correlations between decay at 50 cm and relative elevation (Z*) demonstrate that deeper soil horizons are not isolated from surface processes.

Faster decay rates during the first 3 months at 10 cm and 50 cm were likely driven by leaching from

green and rooibos teas while slower rates in the following nine months may be more representative of microbial decomposition (Duddigan et al., 2020; Lind et al., 2022) (Table 3). The three-month rates may also reflect the summertime deployment since warmer temperatures can accelerate leaching and decay (Kirwan et al., 2014; Lind et al., 2022; Tang et al., 2023). However, seasonality effects are likely small in Georgia where average daily soil temperatures ranged from ~15 to ~ 28 °C between winter and summer (Table 1a).

This is narrower than the temperature gradient in an hours-long leaching study that found ~5% and ~10% increases in green tea mass loss between 8 to 19 °C and 19 to 60 °C, respectively (Lind et al., 2022). Microbial decomposition is temperature sensitive (Yvon-Durocher et al., 2010) but responsiveness in wetlands across latitudes and experiments is mixed (Kirwan et al., 2014; Tang et al., 2023). Seasonal changes in plant production and root-microbe interactions also affect decomposition by altering the belowground chemical

and physical environment (Pett-Ridge et al., 2021; Van Der Nat et al., 1998). Summertime aboveground and rhizome biomasses and stem height correlated positively with decay, particularly in the first six months, indicating that higher plant abundances correspond to increased TBI mass loss (Table 4). It is unclear whether decay responded to plant-microbe interactions or plant effects on soil structure since correlations were inconsistent at twelve-months (i.e., the following summer). We cannot tease apart temperature effects on

leaching and decay further because tea bags were deployed and plants were surveyed only once, during summer. Better assessment of temperature effects on the TBI requires multiple deployments and collections and repeated characterizations of above- and below-ground plant processes across different seasons.



Decay was faster in the top 10 cm, as predicted, but not for the expected reasons (Table 3). We hypothesized that TBI decay and that of green and rooibos teas would be faster in the surface horizon due to

greater rhizodeposition, bioturbation, and more oxidizing conditions. Instead, green tea loss rates were similar at both depths and slower TBI rates at 50 cm were driven by the rooibos tea (Table 3a-b). TBI decay correlated positively with plant characteristics at *both* 10 and 50 cm (Table 4). The rooting zone of *S. alterniflora* extends 20-30 cm and is generally above the 50 cm deployment horizon. Plant effects on soil structure and porewater movement are strongest in the rhizosphere but may extend to deeper depths more

weakly, reflecting the year-over-year soil building process and preservation of dead roots and rhizomes. Burrow density did not consistently correlate with TBI decay, but when it did, relationships were negative meaning that rates slowed with more burrowing, which is contrary to most observations (Table 3) (Kostka et al., 2002a; Kostka et al., 2002b; Xiao et al., 2021). Redox conditions were less negative at 10 cm but were not correlated with TBI decay (Fig. 1; Table 4). This was unexpected given thermodynamic constraints of

anoxia on decomposition, but may be due to loss of compounds that are less redox dependent during the short, one-year incubation whereas decay over longer deployments would become more dependent on processes that are oxygen sensitive, such as depolymerization (Huang et al., 2020; LaCroix et al., 2019). Plant production, bioturbation, redox conditions, and porewater exchange change on time scales of hours-to-seasons and our summertime ecological observations and periodic porewater collections may have been at

too coarse a resolution to adequately capture belowground environmental conditions at 10 and 50 cm. Alternatively, the unexpected correlations (or lack thereof) may highlight that many factors influence decay and that short-term rates are more sensitive to other drivers. Regardless, slower rates at 50 cm demonstrate that decay is more constrained by environmental conditions than the molecular composition of litter, which is consistent with emerging frameworks of organic matter decomposition (Lehmann & Kleber, 2015; Marín-

Spiotta et al., 2014; Spivak et al., 2019).

Differences in TBI decay between 10 cm and 50 cm persisted across the relative elevation (Z*) gradient, with faster rates in plots that were lower in the tidal frame (Fig. 3; Table 5). Marsh surface elevation gradients and tidal flooding dictate many aspects of marsh functioning, including plant production and surface soil stiffness which increased and decreased, respectively, in plots at lower relative elevation (Z*)

levels. Soil temperatures at 10 cm were warmer at lower marsh surface elevations and differed between inundated and flooded tidal stages, but patterns were less clear and seasonally consistent than in a nearby marsh (Fig. 2; Table 1) (Alber & O'Connell, 2019). Porewater exchange is greater at lower elevations and closer to tidal creeks (Guimond & Tamborski, 2021), which can facilitate decomposition by increasing oxygen delivery to the subsurface, removing toxic metabolites, increasing pore-space connectivity, and

altering organic matter-mineral associations (Canfield, 1989; Liu et al., 2008; Xiao et al., 2021). Sharp changes in TBI rates across the relative elevation (Z*) gradient at three months likely reflect more extensive



leaching at lower elevations where soil stiffness is also lower and porewater exchange would be greater (Fig. 3; Tables 3, 4) (Guimond & Tamborski, 2021; Lind et al., 2022). This is also consistent with a sharper drop in TBI rates between 3- and 6- months in plots with the lowest relative elevation (Z\*) values (Fig. 3a, Table 4). More moderate changes in TBI rates with relative elevation (Z\*) at 6- and 12- months indicate that inundation effects on decay extend beyond leaching, which plateaus between 20 (green) and 80 (rooibos) days (Duddigan et al., 2020). Our results contrast with a recent study in Venice Lagoon (Italy) but comparisons are tricky because Puppin et al.'s (2023) analysis combines teabag burial depth (0-24 cm) with marsh surface elevation into the variable $z_\beta$, making it difficult to assess those factors independently. Further analyses focused on the shallowest horizon (8 cm) in Puppin et al. 2023 where TBI k slowed with increasing distance from creekbank edges but showed no correlation with estimated time flooded over the 3-month deployment. In our study, it is not possible to differentiate effects of creekbank distance from flooding duration because they were confounded but testing this could provide insight into the sensitivity of decay to soil structure and hydrology. Because inundation influences many ecological, physical, and biogeochemical factors, we cannot definitively attribute correlations between relative elevation (Z\*) and TBI decay to any single one or a combination. However, because relative elevation (Z\*) was the only variable that consistently correlated with decay at both depths and all three time points, we suspect that gradients in porewater hydrology are particularly important (Table 4). By 12 months the regression slope between TBI decay and relative elevation (Z\*) was ~2x's greater at 10 cm compared to 50 cm, demonstrating that rates become less sensitive to inundation at deeper depths and over time. Moreover, the persistence of correlations between TBI decay and relative elevation (Z\*) at 50 cm shows connectivity between surface and deeper horizons and that environmental conditions below the rhizosphere that affect organic matter loss are not constant or uniform.

Our results suggest that organic matter decay is less sensitive to molecular composition than the soil environment and that porewater hydrology may be a particularly important factor affecting short-term rates. This is largely based on the sensitivity of rooibos tea to soil depth and relative elevation (Z\*) because leaching effects are smaller, compared to green tea, and decay rates are comparable to natural marsh litter (Tables 2-5, S2; Fig. 3). It is possible that kinetic (e.g., temperature) and thermodynamic (e.g., redox) controls become more important over longer timescales, after low molecular weight, soluble compounds are lost and decay is more dependent on depolymerization of larger molecules (Conant et al., 2011; Hu et al., 2020). Deployments beginning in different seasons and lasting longer than one year, and perhaps without green tea, could be useful in assessing within-site sensitivity of decay to temperature and how controls on organic matter loss change over time. Pairing organic matter loss rates with geochemical analyses and rates of porewater



exchange would be valuable to understand molecular-level changes and explore the roles of physicochemical
protection and hydrology.

### 4.4 Conclusions

In this Georgia salt marsh, the TBI produced faster organic matter decay rates relative to studies using local litter. The faster rates were largely due to initial rapid green tea loss and were greatest in the first three months. Placing TBI rates within the context of more traditional approaches is important for assessing the broad applicability of this method and whether changes, such as extending deployment durations and dropping green tea, are warranted. Publishing decay rates of green and rooibos teas alongside the TBI and site-specific literature values is key for evaluating potential method improvements and better identifying generalizable patterns across environmental gradients, such as elevation, flooding, and latitude. We found that rooibos tea produces decay rates comparable to local litter and that rates slow with depth, time, and increasing marsh surface elevation (Tables 2-5, S2; Fig. 3). Because the composition of rooibos tea is similar to natural litter (Duddigan et al., 2020) and preparation is highly standardized, our findings demonstrate that environmental conditions exert stronger controls on decay than molecular recalcitrance, which is in line with current theory (Tables 2-4; Fig. 3) (Lehmann & Kleber, 2015; Marín-Spiotta et al., 2014). Slower, steadier rooibos rates at 50 cm suggest that organic matter surviving transit through the rhizosphere may still be vulnerable to decomposition in deeper, more stable soil horizons. Consistent differences in rooibos decay rates across marsh surface elevation gradients (i.e., Z*), and over time and with depth, indicate that local hydrology strongly affects organic matter loss. This variable is often overlooked in marsh decomposition studies but may be more important than kinetic (e.g., temperature) and thermodynamic (e.g., redox) constraints in the short term.

### Data availability

All raw data have been submitted to the GCE LTER and EDI data archives and will have been assigned a publicly accessible digital object identifier prior to publication.

### Author contributions

The study was designed by S. Reddy, F. Wu, S. Pennings, and A. Spivak. Data collection and sample analyses were performed by S. Reddy, F. Wu, and W. Farrell. S. Reddy and A. Spivak wrote the initial manuscript draft. S. Reddy, F. Wu, W. Farrell, S. Pennings, M. Eagle, J. Sanderman, C. Craft, and A. Spivak contributed to manuscript editing and review.

### Competing interests



The authors declare that they have no conflict of interest.

665 **Acknowledgements**

We thank D. Smith, G. Giordano, S. Dong, R. Lofgren, and A. Pinsonneault for assistance with field deployments and lab analyses and the Georgia Coastal Ecosystem LTER for project support (OCE-9982133). S. Reddy was supported by the University of Georgia's Center for Undergraduate Research. Spivak was supported by NSF DEB-2121019, Georgia Sea Grant, and the US Coastal Research Program. F. Wu was

670 supported by Natural Science Foundation of Fujian Province (2022J05278) and Marine and Fishery Development Special Fund of Xiamen (23YYST064QCB36). Any use of trade, firm or product names is for descriptive purposes only and does not imply endorsement by the U.S. Government. This manuscript is contribution number 1124 from the University of Georgia Marine Institute.



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
