# Peer review of "Decomposing the Tea Bag Index and finding slower organic matter loss rates at higher elevations and deeper soil horizons in a minerogenic salt marsh."

_EGUsphere, 2024_

## Author Response (AR1)

Reviewer 1

We thank Reviewer 1 for their comments and suggestions and feel that these edits have substantially strengthened out manuscript. Below, we include each of the reviewer's comments followed by our response in red font. All line numbers refer to where the edits would be found in a revised manuscript. We hope that these responses have adequately addressed the reviewers questions and concerns.
* * *
Overall, this is an interesting article that investigates the decomposition rate changes of tea bags (TBI) and local plant litter at different soil depths and temperatures, while exploring the factors influencing litter decomposition rates. The writing is generally good, but some sections (such as Results and Discussion) need further refinement. Overall, I suggest major revisions to further enhance the quality of the article. I have several general questions and specific comments as follows:

General Comments:

(1) The title mentions "minerogenic salt marshes." How do these differ from organic marshes? Furthermore, the introduction and discussion sections do not extensively address or explore this distinction. Minerogenic marshes have specific characteristics that could potentially influence the decomposition rates at different soil depths. Could elaborate on this?

The reviewer raises a good point and in response we clarified the characteristics of our study system compared to organic marshes in section 2.1 (study site and design): "Total suspended sediment levels are high in the Altamaha River which feeds into the GCE-LTER domain and contributes to salt marsh vertical accretion (Langston et al. 2021; Mariotti et al. 2024). As a result, these minerogenic marshes have lower soil carbon content, porosity, and permeabilities but higher bulk densities compared to organic-rich marshes (Giblin and Howarth 1984)." (lines 136-140)

We also added the following sentence in section 4.3 of the Discussion to address how differences in soil properties between mineral and organic marshes might affect decay: "Similarly, parallel deployments in mineral and organic marshes with similar flooding regimes but different soil properties (e.g., bulk density, porosity, permeabilities, carbon content, etc.) could provide further insight into how the belowground environment affects decay." (lines 709-712)

(2) In Line 69, the authors discuss plant and animal effects. While the plant effects are covered, how do animals influence decomposition in this context? Specifically, Line 75 emphasizes animal burrows. How might animal burrows impact soil physicochemical properties and soil microorganisms?

To address the reviewers question we added this sentence in section 1 (Introduction): "Bioturbation by animals can strongly affect decay rates in surface soil horizons, by altering redox conditions, digesting organic matter, and moving organic matter between oxic and anoxic layers (Kostka et al. 2002a; Kristensen et al 2012)." (lines 74-76).

(3) In Line 120, the authors mention using local plant detritus. Why is this important? How does comparing this with TBI enhance the study, and what specific questions or problems does it address? This should be clearly explained in the introduction.

Comparing decay rates estimated from the TBI with local plant detritus is important for determining the applicability of this approach to the target study system. Tea used in the TBI represents allochthonous organic matter with a different biochemical composition – and potentially different decay dynamics – than plant detritus produced locally in the wetland. Comparisons of decay rates based on local detritus between marshes can be complicated, however, due to differences in biochemical composition that reflect factors ranging from nutrient availability to plant species identity. The standardized litter approach of the TBI offers a way to evaluate environmental controls on decay without potential confounding factors due to differences in litter organic matter composition. But before applying decay rates estimated with the TBI approach it is important to verify that it produces rates that are comparable to natural local plant detritus.

In response, we added the following clarifying sentences and hypothesis to the introduction (section 1).

"Built into this is the assumption is that decay dynamics of very specific types of terrestrial organic matter (i.e., rooibos and green teas) reasonably approximate those of local plant detritus." (line 109-110)

"Consequently, comparing decay rates estimated from the TBI with natural, local litter is important for determining the applicability of this approach to the target study system." (lines 115-117)

"Lastly, we expected that decay rates of rooibos tea and local plant detritus would be comparable and slower than TBI rates." (lines 129-130)

(4) Lines 128-141 contain detailed information about study locations. Would it be possible to include a map or illustrative figure to better present the experimental setup?

As suggested, an illustrative figure and aerial plot of the study site was added to the section, showing the 3 transects and their respective distance to the main channel and the tidal creek edge.

[Figure]

Figure 1. (a) Study plot orientation relative to the main channel, tidal creek, and relative marsh surface elevation (Z*, see equation 1). Plots were distributed along the tidal creek (C, 0 m), in the marsh platform (P, 14 m), or in between (M, 4 m) across an elevation range of 0.55-1.13 m (NAVD 88). Plot color corresponds to Z* or relative position in the tidal frame (m). Spacing between plots reflects Wu et al.'s (2022) goal of capturing marsh processes around the fan of a headward eroding creek. (b) Aerial photograph of study site from Google Earth with demarcated lines showing approximate plot distribution in respect to the headward eroding creek. Exact plot locations are described by Wu et al. (2022). Plots affected by the headward eroding creek were excluded from this study.

(5) Lines 241-254 raise several questions. The data used are from 2003-2004; could there be discrepancies with the current situation? Also, why use root litter for local litter experiments? The root litter used weighs 10g, whereas TBI uses approximately 1.6g. Does this affect the comparability of the experiments? Moreover, local litter was placed at -10cm and -20cm depths, but not at -50cm. How many replicates were there for TBI, and are they consistent with local litter experiments?

The reviewer is correct to note that the comparison between the 2003-2004 litter deployment and our TBI experiment is imperfect. However, one goal of our work was to assess potential bias of using a non-local organic matter (i.e., tea) to approximate decay of local marsh grass detritus and the dataset we present is the only one – published or not – to provide this comparison. Prior work on decay in this system used source specific isotopes and biomarkers, providing information on a much finer resolution than the bulk, mass loss approach of the TBI. Comparisons between compound class (or compound specific) rates with total mass loss rates can be difficult to interpret because plant litter is comprised of multiple compound classes with different reactivities. Roots deposit organic matter into anoxic soil horizons and this material is more likely to contribute to burial than aboveground shoots and leaves. Consequently, root litter is the best proxy for bulk soil organic matter decay. In response to the reviewers question we added the following clarifying sentence: Root detritus is a good proxy for evaluating soil organic matter dynamics since this material is deposited directly into anoxic horizons and contributes to soil accumulation (lines 284-286).

Both the litterbag technique and our empirical calculations of tea decay are based on relative mass loss. Because of this, it should not matter if the initial mass was 1 or 10 g. Root litterbags were buried between 10 and 20 cm, which is slightly deeper than the shallow tea bags (10 cm). Because the rhizosphere extends to ~30 cm we believe that the slight difference in burial depth between the root litterbags and the shallow tea bags should not affect decay rates. Recognizing that environmental conditions that could affect decay differ between surface and deeper horizons we only compared the root litterbag rates with the shallow (10 cm) horizon tea bags (e.g., lines 518-520: "... TBI rates at three months are 2.8 – 7.2 times faster than prior studies but that drops to roughly double over longer, 6-12 month periods (10 cm horizon only...")

To address the concern about differences in environmental conditions between the litterbag experiment and the teabag deployments we compared monthly mean temperatures and precipitation between 2003 and 2019. This is reported at lines 289-291: "Environmental conditions including precipitation and temperature were similar (p > 0.05) between the litterbag study in the

summer of 2003-2004 (7.7±1.4 cm yr$^{-1}$ and 20.4±2.1 °C) and TBI experiment in the summer of 2019-2020 (12.1±1.9 cm yr$^{-1}$ and 21.4± 2.3 °C).”

The goal of this aspect of the manuscript was to evaluate whether the TBI produces decay rates comparable to natural litter and we believe that slight differences in deployment depth and environmental conditions cannot account for the 120 % difference (TBI vs. natural litter) and 36 % difference (rooibos tea vs. natural litter) between these two approaches.

There were more replicates in our TBI experiment (23 per depth and time point across 3 transects) than in the litterbag experiment (4 reps per time point in the marsh levee and plain sites).

(6) Lines 255-293, Data Analysis, is overly detailed and needs to be condensed for clarity.

As suggested, we have made the Data Analysis section more succinct. The section now reads as follows at lines 294-329:

“Changes in belowground environmental conditions across marsh surface elevations, between soil depths, and over time were assessed with regression analyses and t-tests. Distance category identity (C, M, P; Fig. 1) was excluded from all statistical analyses because it was confounded with marsh elevation, which we predicted would be a key factor affecting environmental conditions and decay rates. Tidal flooding effects on soil porewater chemistry and temperature were tested by constructing regression models against relative elevation (Z*). Porewater data were then aggregated by sampling event and two sample t-tests were used to detect differences between 10 cm and 50 cm depths. Correlations between Z* and soil temperature were further tested by partitioning according to season (summer: 18/7/2019 to 22/9/2019; fall: 23/9/2019 to 22/12/2019; winter: 23/12/2019 to 19/1/ 2020) and periods of tidal inundation or exposure; differences between slope coefficients were evaluated based on Clogg et al., (1995). We tested whether soil temperatures differed between depths within each season using paired t-tests.

We tested whether decay rates (TBI k, empirical $k_g$, empirical $k_r$; d$^{-1}$) and stabilization factors (TBI S, empirical $S_g$, empirical $S_r$) differed over time (3-, 6-, or 12-months) and between soil depths (10 vs 50 cm) by constructing linear mixed effect models using the nlme package for R (Pinheiro et al., 2016). The mixed models evaluated the fixed effects of time and depth and included plot number as a random factor. We then conducted paired t-tests to further explore how TBI and empirical decay rates, decomposable fractions (TBI and empirical $a_g$ and $a_r$), and stabilization factors changed over time within a depth horizon.

Potential drivers of TBI k were evaluated by calculating Spearman rank correlation coefficients between rates and environmental conditions for the three time points (3, 6, or 12 months) and two soil depths (10 and 50 cm). Porewater data for the 3, 6, and 12 month periods were combined with data from previous time points (e.g., 2, 3, or 6 months) to better represent cumulative conditions. Temperature was excluded because the shared time series with TBI k violated assumptions of independence. The TBI k values correlated strongly with relative elevation (Z*), *S. alterniflora* rhizome and aboveground biomass, and soil stiffness. We then evaluated interdependencies between these potential drivers by using subsequent single-factor regressions of relative elevation (Z*) vs. aboveground biomass and soil stiffness. Correlations between these variables limited further hypothesis testing of decay drivers to plot position within the tidal frame (Z*). We tested whether TBI k rates and S values changed with relative elevation (Z*) using linear regression models and then evaluated differences between the resulting slope coefficients over time and with depth, as described by Clogg et al., (1995).

Data were tested for outliers using a 1.5 interquartile range cutoff and $\log_{10}$ transformed as needed to meet assumptions of normality. Analyses were conducted using R software (R Development Core Team, 2023. Data are presented as means ± standard error (SE) unless noted otherwise."

(7) The results section presents extensive data and comparisons, thoroughly examining the data. However, this section should focus more on objectively describing data changes and significant differences, avoiding excessive interpretation and discussion. For instance, Lines 342-344, 370-375, 388-390, and 396-397 include speculative comments that are better suited for the discussion section.

Those sentences were removed from the results section, as suggested.

(8) In the results section, the authors used a non-traditional TBI index calculation, separately calculating the decomposition rate and stabilization factor for green tea and rooibos tea. However, the terminology must be consistent throughout (e.g., kg, kr, Sg, Sr). Phrases like S=Sg and TBI decay appear in the text. The same issue exists in the discussion section. Please ensure consistent terminology.

In response, we further clarified terminology used to distinguish the TBI and empirical approaches (copied below) and thoroughly edited the manuscript to ensure consistency throughout.

"From here forward, rates and variables calculated using Keuskamp et al. (2013) or the first order decay approach are referred to TBI and empirical, respectively. For the TBI, this includes TBI k (eq. 3), TBI $a_g$ (fraction mass loss of green tea), TBI $a_r$ (eq. 5), and TBI S (eq. 4) to denote decay rate, the decomposable fractions of green and rooibos teas, and the stabilization factor, respectively. The empirical calculations include $k_g$ and $k_r$ (eq. 6), $a_g$ and $a_r$ (mass fractions lost of each tea type), and $S_g$ (eq. 4) and $S_r$ (eq. 7) where g and r refer to green or rooibos teas, respectively. Importantly, there are two commonalities between these approaches: TBI $a_g$ is the same as empirical $a_g$, and TBI S is the same as empirical $S_g$." (lines 267-274)

(9) Lines 509-525 commendably summarize the discussion on TBI results and the relationship between k and S. However, the discussion would benefit from a clearer connection to the study's findings and implications.

We appreciate the reviewers positive feedback and in response have made the connections to our study clearer. The goal of this section of the manuscript was to contextualize our TBI decay rates relative to natural litter decay in the same study system and TBI rates in wetlands globally. These comparisons are important for interpretation and extrapolation of TBI rates as well as for using the main strength of the TBI approach – uniformity of litter composition – to test large scale ecological theories, such as the Metabolic Theory of Ecology.

Lines 530-537: We expected TBI rates to be faster in our study in subtropical Georgia, USA, compared to higher latitudes based on the metabolic theory of ecology, which predicts that decay rates increase with rising temperatures (Yvon-Durocher et al., 2010), and observations that warming accelerates loss of labile compounds in soils (Conant et al., 2011; Melillo et al., 2002). To test this, we compared our rates with those from 7 other salt marsh TBI studies that encompass 11 countries and span a latitudinal gradient of 93.7° (-37.7° to 56°; Fig. 5, SI table 2) (Mueller et al.,

2018, Puppin et al., 2023, Marley et al., 2019, Alsafran et al., 2017, Yousefi Lalimi et al., 2018, Sanderman and Eagle, unpub, Tang et al., 2023).

Lines 546-548: "Our Sapelo Island, GA, 3-month rates were similar to salt marshes in North Carolina, Virginia, Maryland, California, and Massachusetts, USA and Zeijhong Province (ZJ), China (Fig. 5)."

Lines 568-573: "In our experiment on Sapelo Island (GA, USA), the highest S values coincided with the fastest decay rates in the first 3 months (Fig. 5). Stabilization values then decreased between 3- and 6- months but there was no overall temporal trend because values increased at 12-months. This variability is not unique to our site; TBI S values decreased and increased at East Lothian (ELN), Scotland and Schleswig-Holstein (SH), Germany, respectively, between 3- and 12- months (Fig. 5)."

Lines 579-581: "This interpretation, though, is caveated our experimental findings and by Mori et al.'s (2022) discussion that green and rooibos teas do not share S values which violates assumptions of the TBI method (Table 3)."

Lines 582-584: "Faster decay rates estimated in this study using the TBI method relative to more conventional litterbag and laboratory experiments suggest that these approaches are not interchangeable."

Lines 590-592: "Few studies like ours have directly compared decay rates from the TBI, its components, and more conventional approaches but this would be useful in assessing whether Keuskamp et al.'s (2013) method can be applied broadly, in dry and saturated soils"

Specific Comments:

(1) Line 32: Replace "Tea BI rate" with "TBI."

As suggested, we replaced "Tea BI" with "TBI".

(2) Line 59: The authors mention that effects on soil organic matter decay are "less well understood." What specific scientific questions or reasons contribute to this lack of understanding? Please clarify.

This sentence was deleting during editing of the introduction.

(3) Line 102: When discussing the advantages of TBI, "inexpensive" lacks professionalism. Consider using a more precise term.

In response, we have changed the term "inexpensive" to "cost effective".

(4) Line 137: Why were tea bags placed at -50 cm, and how was soil disturbance minimized during placement? How many experimental replicates were there?

Tea bags were placed at 50 cm to see how being in a more stable environment with fewer effects from aboveground abiotic and biotic variables (plant roots, crab burrows, tidal flooding) could affect decomposition. We added the following clarifying sentence at lines 222-223: "Burial depths were chosen to assess decay rates within the more dynamic rhizosphere (10 cm) compared to more stable, deeper horizons (50 cm)."

Disturbance at the time of deployment was minimized by inserting a trenching shovel to 50 cm depth and pushing it forward ~5 cm to create a small wedge of space in the soil column. Then a PVC pole with green and rooibos teabags attached with cable ties at 10 and 50 cm was placed into the soil. Because no material was removed, the soil column squeezed back together following shovel removal. Three replicate poles and attached tea bags were placed at each plot and collected at different time periods (3, 6, or 12 months). Recognizing that these actions disrupt soil porewaters and redox conditions we did not collect samples for 2 (porewater) or 3 (tea bags) months.

The experimental design is described at lines 141-144: "Study plots were established in summer 2019 along a tidal creek, with a total of 23 plots placed at 3 distances from the creekbank edge (creek: 0 m, 7 plots; middle: 4 m, 8 plots; platform: 14 m, 8 plots) that captured a range of marsh surface elevations, from 0.55 to 1.13 m (North American Vertical Datum of 1988, NAVD 88; Fig. 1)."

Teabag deployment is described at lines 220-222: "Tea bags were dried at 60 °C to constant mass prior to deployment, during which triplicate bags of each tea type were buried in every plot at 10 and 50 cm depth in July 2019 (initial tea weights: rooibos: 2.01±0.004 g; green: 2.17±0.004 g)."

Consequently for each time point there were 23 replicates at 10 and 50 cm depth, distributed across the 3 distance categories from the creekbank edge (creek, middle, and platform).

(5) Line 208: Specify the initial weight of the tea bags.

We added the initial weights of the tea bags, "Rooibos: 2.01±0.004 g; Green: 2.17±0.004 g" at line 222.

(6) Line 228-229: Why does the methods section introduce potential results and discussion points instead of presenting them in the results section?

We included these sentences as justification for the following description of model selection. Because this information is necessary to understand the data analyses we feel that it is important to include in section 2.6. For these reasons no changes were made.

(7) Line 195: How were the crabs and snails measured or quantified?

In response, we added this information: "The densities of burrows (>0.5 cm diameter, all species pooled) and snails (>0.3 cm spire height) were recorded in 0.5 × 0.5 m quadrats at each plot as individuals m$^{-2}$." (Lines 210-211)

(8) Line 288: Correct the citation format to "Clogg et al., (2009)."

As suggested, the citation formatting was corrected.

(9) Previous comments may have already addressed this, but I remain curious. This study focuses on minerogenic marshes. Are the articles discussed in the discussion section based on minerogenic marshes, or do they include other types of wetlands as well?

Few studies distinguish between minerogenic and organic marshes. We only referenced other salt marsh studies but because of this reporting gap we did not differentiate according to whether they were minerogenic or organic. However, we feel that knowing our study site is minerogenic is

important for interpreting and extrapolating the results. The changes made to the text in response to this comment are the same as those for Reviewer 1's first general comment (above).

Reviewer 2.

We thank Reviewer 2 for their comments and suggestions and feel that these edits have substantially strengthened out manuscript. Below, we include each of the reviewer's comments followed by our response in red font. All line numbers refer to where the edits would be found in a revised manuscript. We hope that these responses have adequately addressed the reviewers questions and concerns.
* * *
The study is commendable for its attempt to delve into the potential weaknesses of the TBI method, such as exploring the effect of leaching and the assumption that the stabilization factor remains constant across different organic materials. Additionally, the study examines the influence of specific physical and biotic characteristics on decomposition, although distinguishing the causal relationships between the various drivers presents a challenge.

The writing is generally solid, but certain sections require additional polishing. Overall, I suggest major revisions to improve the quality of the article. In particular, I suggest improving the clarity of the method presentation and discussion in certain areas and better distinguishing between methods, results, and discussion. It's important to avoid anticipating results when explaining methods and to refrain from including discussion points when objectively presenting results.

We thank the reviewer for this feedback and suggestions to improve our manuscript.

General Comments:

In the Methods section, the authors thoroughly describe their experimental design, including site locations, depths, and the various variables measured. To enhance clarity, would it be possible to add a figure featuring an image of the study site and a schematic of the experiment? Given the complexity and detail of the experiment, this would help the reader more clearly and directly understand the setup and more easily follow the results later on.

As suggested, we added a figure showing plot distributions relative to the main channel and the tidal creek edge and relative positions in the tidal frame (Z*).

[Figure]

Figure 1.  (a) Study plot orientation relative to the main channel, tidal creek, and relative marsh surface elevation (Z*, see equation 1). Plots were distributed along the tidal creek (C, 0 m), in the marsh platform (P, 14 m), or in between (M, 4 m) across an elevation range of 0.55-1.13 m (NAVD 88). Plot color corresponds to Z* or relative position in the tidal frame (m). Spacing between plots reflects Wu et al.'s (2022) goal of capturing marsh processes around the fan of a headward eroding creek. (b) Aerial photograph of study site with demarcated lines showing approximate plot distribution in respect to the headward eroding creek. Exact plot locations are described by Wu et al. (2022). Plots affected by the headward eroding creek were excluded from this study.

The Methods section is extensive and detailed. However, in several instances, the authors anticipate and comment on results (lines 169, 186-187, 189-190, 195-196, 228-230, 281-283). Even though some of these results are from another study (Wu et al., 2022), it may be more appropriate to present and discuss them in the Results and Discussion sections, given that they are used as variables in correlation analyses.

We deleted the requested lines from the methods section and added a short paragraph in the results section (3.3) that describes relevant findings from Wu et al. (2022).

"Plant characteristics, animal abundances, and soil stiffness were reported previously by Wu et al., 2022. Soil shear strength was measured in the top 4 cm using a field shear vane (GEONOR H-6O). Spartina alterniflora aboveground biomass was estimated based on stem density counts and known masses of representative stems. Belowground biomass was measured by collecting soil cores (10 cm diameter, 30 cm depth) centered on a culm of S. alterniflora in each plot and then washing roots and rhizomes free of soil before drying and weighing. Two major groups of invertebrates were present: crabs (Uca pugilator, Sesarma reticulatum, Panopeus) and snails (Littoraria irrorata). The densities of burrows (>0.5 cm diameter, all species pooled) and snails (>0.3 cm spire height) were recorded in 0.5 × 0.5 m quadrats at each plot as individuals m-2." (lines 203-211)

Considering that the study provides an interesting analysis of the weaknesses of the TBI method, I suggest further clarifying and expanding the description of Keuskamp et al. (2013)'s method. At line 214, it is important to specify that the TBI k coefficient represents the decomposition rate constant of the labile fraction. According to Keuskamp et al. (2013), the decomposition rate constants for the labile and recalcitrant fractions are denoted by k1 and k2, respectively. During the initial phase, the labile fraction is rapidly decomposed, and the weight loss of the litter is primarily determined by k1. Additionally, in section 2.6, as discussed later, it should be specified that, according to Keuskamp et al. (2013), S is assumed to be equal for both tea types, meaning that the environmental stabilization of the labile material is considered independent of the relative size and composition of the hydrolysable fraction. The TBI method relies on several assumptions since its purpose is to measure k and S without requiring time series data by using two types of tea with differing characteristics. Clarifying this point would help the reader better follow the discussion later on.

We thank the reviewer for this clarification and note that we describe the 2 decay coefficients, albeit in slightly different terms in section 1 (lines 105-108): "One key assumption is that the decay dynamics and chemical composition of two different litter types (green and rooibos teas) can be integrated to estimate loss of natural detritus that has characteristics of the proxy constituents. The TBI, in effect, applies a simplified two-pool decay model and assumes that the biochemical compositions of both pools are broadly applicable."

In response to the reviewers suggestion we added equation 2 ($W(t) = ae^{(-k_1 t)} + (1-a)e^{(-k_2 t)}$, line 232) which describes a 2-pool model and explain the modifications made by Keuskamp et al to arrive at the TBI approach.

"Equation 2 combines decay of labile (k1) and refractory (k2) organic matter and requires time series data. The TBI eliminates the need for a time series by simplifying equation 2 to equation 3 using the assumptions that decay rates of refractory organic matter are negligible (i.e., k2 = 0) and that the decomposable fraction of organic matter (i.e., a) can be represented by combining different characteristics of rooibos and green teas. In equation 3, W(t) is the mass fraction of rooibos tea remaining at time t, k is the decay coefficient, and S is a stabilization factor. The inhibitory effect of environmental conditions on decay (i.e., S) is calculated based on green tea but assumed to be the same for both tea types. The decomposable fraction (a) of green tea (ag) is estimated by the mass fraction lost while that of rooibos tea (ar) is based on its hydrolysable fraction (Hr) and S. We used the tea-specific H values reported by Keuskamp et al. (2013) that were calculated as the sum of nonpolar extractable, water soluble, and acid soluble fractions (Hr: rooibos, 0.552 g g−1; Hg: green, 0.842 g g−1)" L236-247

At the end of paragraph 2.6, the authors mention modifying the S equation for rooibos tea (Sr) by substituting ar and Hr, but they do not specify how this calculation was performed. Could the authors clarify how they calculated Sr? Specifically, how were the values of ar and Hr obtained? How did the authors determine when all the labile material in the rooibos tea had decomposed?

We added the modified Sr equation (7) at line 266: $S_r = 1 - \frac{a_r}{H_r}$.

Here, $H_r$ is the hydrolysable fraction of rooibos tea and $a_r$ is the decomposable fraction based on rooibos tea mass loss during the experiment.

At lines 246-247, the authors mention comparing TBI results with a prior litterbag experiment conducted from June 2003 to 2004. Did the authors evaluate the environmental conditions (e.g., temperature, rainfall) of the 2003-2004 period to determine if they are comparable to those of the TBI experiment?

To address this question, we compared monthly mean temperatures and precipitation between 2003 and 2019. We now report that "Environmental conditions including precipitation and temperature were similar (p>0.05) between the litterbag study in the summer of 2003-2004 (7.7±1.4 cm yr$^{-1}$ and 20.4±2.1 °C) and TBI experiment in the summer of 2019-2020 (12.1±1.9 cm yr$^{-1}$ and 21.4± 2.3 °C)." (lines 289-291)

In several captions of tables (Table 1, Table 3, Table 5), the authors state, "Significant differences (<0.05) are denoted by superscripts". I would suggest specifying which statistical test was used to determine the significance of the differences and clarifying that different letters indicate significant differences.

In response, we added information about the type of test(s) used in the descriptions for Tables 1, 3 and 5, and include references to section 2.8.

The Results section is extensive and detailed. However, in several instances, the authors anticipate interpretation and discussion (lines 326-328, 342-344, 366-367, 388-390,396-397). The Results section would benefit from focusing more on objectively presenting the results, with speculative comments reserved for the Discussion section.

As suggested, these lines were deleted from the results section.

Specific Comments:

Lines 63-68: In the Introduction, the authors state that "water passage through the subsurface alters the thermodynamic favorability of different pathways for decomposition" without explaining how this occurs. Immediately after, they add that "the intensity of tidal flooding effects on plant and soil processes is strongest at creekbanks and lower elevations relative to interior and higher elevation areas," but they do not clarify the reasoning behind this statement. To enhance comprehension for a multidisciplinary audience, I suggest briefly explaining these assertions.

We appreciate this feedback and in response edited and rearranged this paragraph, which now reads as:

"Tidal flooding effects on plant and soil processes are generally strongest along creekbanks and at lower elevations, which are inundated more frequently and for longer relative to interior and higher elevation areas (Guimond & Tamborski, 2021; Howes & Goehringer, 1994; Reed & Cahoon, 1992). Rising and falling tides result in oscillating soil redox conditions, with greater oxygen penetration during emergent periods and more strongly reducing conditions under submergence (Fettrow et al., 2023; Seyfferth et al., 2020; Spivak et al., 2023). Flooding changes the soil environment for decomposition by altering availability of terminal electron acceptors, increasing pore space connectivity, leaching organic matter, and changing microbial access to bioavailable compounds

(e.g., sorption, enzyme functionality, molecular configuration) (Bradley & Morris, 1990; Giblin & Howarth, 1984; Liu & Lee, 2006; Morrissey et al., 2014)." (lines 57-66)

Line 95: I suggest including a couple of examples of geochemical approaches for the sake of clarity.

As suggested, we added examples of geochemical tools that have been used to trace organic matter sources, transformations and fates in coastal wetland ecosystems at lines 95-100.

"Geochemical approaches describe organic matter loss and transformations (e.g., C content, stable isotopes, thermal reactivity, biomarkers), can be applied over timescales of seasons to centuries (e.g., radiocarbon), and benefit from multiple proxies, but are resource intensive and require specialized instrumentation (e.g., mass spectrometry) (Benner et al. 1984a, Benner et al. 1984b; Benner et al., 1987; Benner et al., 1991; Duddigan et al., 2020; Luk et al., 2023; Luk et al., 2021; Moran et al., 1989)"

Line 102: I would suggest using a different term instead of "inexpensive" to describe the method's advantage, as it is likely more accurate to say it is less expensive than other methods.

As suggested, the term "inexpensive" was changed for "cost effective".

Line 121-125: The last lines of the introduction seem to anticipate some discussion of the results. However, I believe the authors may be aiming to explain the rationale behind their experimental design. If that's the case, I suggest rephrasing the paragraph to make this intention clearer.

In response this comment and one above we have rewritten the last paragraph of the introduction to focus on our hypotheses.

"We predicted that TBI decay rates would be fastest in shallower soil horizons and lower marsh elevations, where porewater flushing is greater and more frequent, and positively correlated with plant biomass, bioturbation (as crab burrow density), and porewater redox levels. We further predicted that TBI rates would be fastest in the first three months and then decrease over the following nine months, and that this pattern would be more pronounced in shallower compared to deeper horizons. Lastly, we expected that decay rates of rooibos tea and local detritus would be comparable and slower than TBI rates." (lines 124-130)

Line 135: As this is the first time this acronym NAVD appears in the text, I suggest spelling out its meaning, as done in the following lines (line 146: North American Vertical Datum of 1988 (m, NAVD 88)).

We thank the reviewer for catching this error and have made the correction.

Lines 165-173: This paragraph starts with a description of the method used to measure porewater chemistry before listing the measured variables. For clarity, I suggest stating which variables were analysed at the beginning of the paragraph.

As suggested, we added a sentence specifying which variables were measured at the beginning of the paragraph on lines 185-187 that reads: "We collected samples for porewater salinity, pH, and redox in each plot using passive sippers with collection windows at 10 and 50 cm that were deployed in July 2019 (Hughes et al 2012, Paludan and Morris 1999)."

Line 203: Are the Lipton™ tea bags used in this study the same type as those used by Keuskamp et al. (2013)?

The tea bags used in our experiment are the same as those used by Keuskamp et al. (2013). In response we edited the following sentence at 216-219 to read "This method, introduced by Keuksamp et al., (2013), assumes that natural litter is comprised of labile and refractory pools that can be represented by Lipton™ green (European Article Number: 87 22700 05552 5) and rooibos (European Article Number: 87 22700 18843 8) teas, respectively, which we used here."

Line 218: I would specify that Hg and Hr were measured by Keuskamp et al. (2013) using a sequential extraction technique, with the hydrolysable fraction defined as the sum of nonpolar extractives (NPE), water solubles (WS), and acid solubles (AS), as opposed to the recalcitrant nonhydrolysable fraction, which includes AIS and ash.

We included this clarification, as requested. "We used the tea-specific H values reported by Keusksamp et al. (2013) that were calculated as the sum of nonpolar extractable, water soluble, and acid soluble fractions (Hr: rooibos, 0.552 g g$^{-1}$; Hg: green, 0.842 g g$^{-1}$)." (lines 245-247)

Line 290: When stating that data were transformed as needed to meet assumptions of normality, it would be helpful to mention specific examples of transformations used (e.g., log10).

We have specified that a log10 transformation was used to meet assumptions of normality.

Line 297: The authors state that porewater chemistry was surprisingly insensitive to relative marsh elevation. However, immediately after, they mention that salinity, which was previously listed among the porewater chemistry variables, positively correlates with relative elevation. Given this positive correlation, it would be more accurate to say that porewater chemistry is weakly sensitive to elevation.

As suggested, we changed "surprisingly insensitive" to "weakly sensitive" at line 334.

Figure 1: To present the data more clearly, it might be helpful to indicate the number of samples that constitute each box in the box plots.

The number of samples for each box ranged from 20-23, which reflects an inability to collect water or sample loss. We added a statement in the Figure 2 caption that clarifies sample number but omitted those numbers from the figure because including them cluttered the figure. The first sentence of the figure caption was changed to: "Soil porewater salinity (a), redox (b), and pH (c) pooled across all 23 study plots at 2, 3, 6, and 12 months (September, October, January, and July, respectively) at 10 cm (red) and 50 cm (blue) depth (n=20-23 for each box at each depth)." (lines 351-354)

Table 1: The caption states that "Significant differences between tidal stages within a season are denoted by * (p<0.05)." However, it is unclear what the double asterisk (**) signifies in the table.

We edited the Table 1 description to signify that ** corresponds to a p-value < 0.01 on the description for Table 1.

Line 319: As this subsection presents both decay rates and stabilization factors, it would be more appropriate to include "stabilization factors" in the title, rather than mentioning only decay rates.

As suggested, Stabilization Factors has been added to the section title on line 357.

Line 320: For completeness of the results, it would be helpful to mention the values of k and S calculated, as is done at line 346 for the litterbags experiment.

Due to the varied TBI and empirical decay rates and stabilization factors calculated at multiple depths (Table 3), we felt that it would be repetitive and cumbersome to list these numbers again in the text. We added references to Table 3 where the full suite of k and S values can be found and easily compared (lines 358, 364, 368, and 376).

Figure 3: The caption states that "Contrasts between k and S at the 3- (blue), 6- (red), and 12- (green) time points are denoted by letters of the same color." However, it is unclear what "contrasts" refers to. I suggest to clarify the meaning of "contrasts" and explain how the letters are used to denote differences.

The figure description has now been changed to the following for clarity on lines 387-394:

Figure 4. TBI decay rates (a, b) and TBI stabilization factors (c, d) and empirical decay rates (e, f, g, h) at 10 cm (left) and 50 cm (right) and at 3-, 6-, or 12- months (blue, red, and green, respectively). Decay rates decreased while TBI stabilization factors increased with relative marsh surface elevation within the tidal frame (Z*) at both 10 cm and 50 cm depths. Significant correlations (p<0.05, as determined by linear regressions, see section 2.8) are denoted with solid lines. Significant differences between linear regressions of TBI k, empirical kg and kr, and TBI S with Z* at the different time points are denoted by letters of the same color. See Table 2 for statistical results

Table 4: The table would be clearer if it indicated that the numbers presented are Spearman's rank correlation coefficients (ρ).

As requested, the description of the Table 4, has been changed to "Spearman rank correlation coefficients between TBI decay rates (TBI k, d$^{-1}$) and potential abiotic and biotic drivers at 10 and 50 cm depth and for the three deployment intervals (3, 6, or 12 months). Significant correlation coefficients (p < 0.05) are denoted by *."

Lines 403-404: Although it is useful and interesting to calculate both TBI k and tea-specific rates to compare them with previous findings and assess their behaviour with respect to environmental variables, I am unsure how accurate it is to directly compare these values, as they result from different calculations. If they are compared, I suggest providing commentary on the differences in their calculations.

We appreciate the reviewers feedback but respectfully disagree that the comparisons are inappropriate. As we note in the introduction, there is considerable "variability in soil carbon stocks and accumulation rates within and across marshes (lines 46-47).... Characterizing patterns in organic matter decay across tidal inundation gradients and soil depths may therefore provide a useful framework to assess processes contributing to marsh-scale spatial variability in carbon stocks." (lines 54-56) Many studies share the objective of measuring organic matter decay rates with the larger goals of addressing this spatial variability, identifying controls on decay, and parameterizing ecosystem- to global-scale models of carbon cycling. Studies have employed a wide range of techniques to estimate decay rates and the TBI is another, more recent tool. Since the

goal is to understand controls on decay we believe method intercomparison is vitally important. For instance we note that "[p]lacing TBI rates within the context of more traditional approaches is important for assessing the broad applicability of this method..." (lines 718-719).

In the introduction we describe some of the more traditional approaches as well as their strengths and caveats relative to the TBI. We also conclude section 4.2 noting that "Faster decay rates estimated in this study using the TBI method relative to more conventional litterbag and laboratory experiments suggest that these approaches are not interchangeable... Similar decay rates between rooibos tea and more conventional approaches suggest that this aspect of the TBI could be a reasonable proxy when the experimental goal is to assess drivers independently of site-specific differences in organic matter composition and material preparation. Few studies like ours have directly compared decay rates from the TBI, its components, and more conventional approaches but this would be useful in assessing whether Keuskamp et al.'s (2013) method can be applied broadly, in dry and saturated soils." (lines 582-592).

Lastly, we do provide a comparison of rate calculation approaches in equations 2, 3, and 6. To increase comparability amongst the approaches used here we used equation 6 for the tea-specific rates and the litterbags rates.

Lines 447-448: Since green tea decomposition is used to determine ag, which is then used to calculate S, and from which ar is derived to subsequently determine k from W(t)r and ar, does the TBI k coefficient by Keuskamp et al. (2013) represent the decomposition rate of rooibos tea?

We appreciate the reviewer's confusion and agree that the Keuskamp equations are convoluted. In short, the TBI does not directly represent decay of rooibos tea. The decay rate of rooibos tea would be significantly slower if the labile fraction of the green tea was not considered in the calculation.

To address this question and a similar one from reviewer 1 we added more detail to the section describing the Keuskamp equations at lines 231-247:
"Decay rates were calculated per Keuskamp et al. (2013) using four equations:

$$W(t) = ae^{(-k_1 t)} + (1-a)e^{(-k_2 t)}, \quad (2)$$

$$W(t) = a_r e^{(-kt)} + (1-a_r), \quad (3)$$

$$S = 1 - \frac{a_g}{H_g}, \quad (4)$$

$$a_r = H_r(1-S). \quad (5)$$

Equation 2 combines decay of labile ($k_1$) and refractory ($k_2$) organic matter and requires time series data. The TBI eliminates the need for a time series by simplifying equation 2 to equation 3 using the assumptions that decay rates of refractory organic matter are negligible (i.e., $k_2$ = 0) and that the decomposable fraction of organic matter (i.e., $a$) can be represented by combining different characteristics of rooibos and green teas. In equation 3, $W(t)$ is the mass fraction of rooibos tea remaining at time $t$, $k$ is the decay coefficient, and $S$ is a stabilization factor. The inhibitory effect of environmental conditions on decay (i.e., $S$) is calculated based on green tea but assumed to be the same for both tea types. The decomposable fraction ($a$) of green tea ($a_g$) is estimated by the mass fraction lost while that of rooibos tea ($a_r$) is based on its hydrolysable fraction ($H_r$) and $S$. We used the tea-specific $H$ values reported by Keuskamp et al. (2013) that were calculated as the sum of

nonpolar extractable, water soluble, and acid soluble fractions ($H_r$: rooibos, 0.552 g g$^{-1}$; $H_g$: green, 0.842 g g$^{-1}$).

Lines 462-471: In these lines, the authors compare TBI decay rates with measurements from previous studies in the same area. However, they note that these earlier measurements were based on lignin, losses of structural polysaccharides, or plant tissue mass. I am unsure how accurate it is to directly compare these values, as they result from different calculations.

We agree with the reviewer that comparisons between rates based on mass loss vs. compound classes (or specific compounds) can be difficult to interpret. Recognizing this, we included the following information to provide context for interpretation: "The slowest rates were based on lignin while faster rates were estimated from losses of structural polysaccharides (cellulose, hemicellulose) or plant tissue mass. The highest decay rate was calculated from polysaccharides in root and rhizome litter (Benner et al., 1991) and was 30-73% faster than root mass loss along creekbank levees (0.0015 d$^{-1}$) and marsh interiors (0.0020 d$^{-1}$; Table S2). The TBI rates at three months were 2.8 – 7.2 times faster than prior studies but that drops to roughly double over longer, 6-12 month periods (10 cm horizon only; Table 3; Fig 4), with the exception of the rapid polysaccharide-specific rate (Benner et al., 1991). This is perhaps not surprising since nuclear magnetic resonance (NMR) spectroscopy demonstrates sharp reductions in O-alkyl compounds consistent with carbohydrates and polysaccharides and aromatic compounds consistent with tannins during green tea incubations (Duddigan et al., 2020)." (lines 514-524).

Importantly the compound class and compound-specific rates described above are the only published rates available for our study system. We felt it important to include these rates, despite the imperfect comparison, because local site conditions (like hydrology) can strongly affect organic matter decay. In general, decay rates reported by other studies in our system (Benner et al., 1984; Benner et al. 1987; Benner et al., 1991; Rice & Tenore, 1981) were calculated using first order exponential decay models, which is the same as equation 6 in our manuscript which we used to estimate tea-specific and litter bag decay rates.

Figure 4: I suggest mentioning in the caption what the bars represent (e.g., standard deviation). Additionally, would it be possible to include standard deviation bars for the other studies presented in the figure?

As suggested, the phrase "error bars represent standard error" was added to the figure caption. Due to data availability it was not possible to calculate standard error for all of the studies in this figure. We agree with the reviewer that it would be preferable to include error estimates for all points in this plot.

Lines 619-620: The authors mention that their results suggest organic matter decay is less sensitive to molecular composition than to the soil environment. To strengthen the discussion, I suggest briefly elaborating on the reasons supporting this assertion.

In response we added the following sentences at 698-703: "Tea composition is highly standardized but decay rates were faster at lower relative marsh elevations (Z*) and shallower soil depths (10 cm), which points to the importance of environmental controls such as tidal flushing on organic matter loss. Rooibos tea decay rates were slightly higher, but comparable, to natural marsh litter

and less affected by leaching than green tea, and may therefore be a reasonable proxy for organic matter breakdown over certain timescales (Tables 2-5, S2; Fig. 4)."

Technical corrections:

Line 41: At the end of the abstract, there's a typographical error with a phrase that remains incomplete.

The phrase has been deleted, as its inclusion was a typographical error.

Line 265: The punctuation is missing.

The punctuation on this line has been corrected by adding a period.

Figure 4: Figure 4 is abbreviated as "Fig. 4," unlike the other figures. For consistency, I suggest standardizing the abbreviation format across all figures.

As per suggestion, we have changed the "Fig. 4" label to "Figure. 4" to match the other plots.

---

## Author Response (AR2)

Reviewer 1

We thank Reviewer 1 for their comments and suggestions.
* * *
Reviewer 2

We thank Reviewer 2 for their comments and suggestions and feel that these edits have substantially strengthened out manuscript. Below, we include each of the reviewer's comments followed by our response in red font. All line numbers refer to where the edits would be found in a revised manuscript. We hope that these responses have adequately addressed the reviewers questions and concerns.

Specific Comments:
I only noticed a minor typographical error at line 150 (of the ATC1 version) ('Built into this is the assumption is that…'), which should be corrected.

In response, we have corrected the line to read "An assumption of this model is that the decay dynamics…." on line 109 of the revised manuscript.
* * *
Editor

We thank the Editor for their comments and suggestions and feel that these edits have substantially strengthened out manuscript. Below, we include each of the editor's comments followed by our response in red font. All line numbers refer to where the edits would be found in a revised manuscript. We hope that these responses have adequately addressed the editor's questions and concerns.

Referring to Figure 1b: With the next file upload request, please include the copyright icon as follows: © Google Earth

In response, we have corrected the line to read "Aerial photograph of study site (© Google Earth 2024) with demarcated lines showing approximate plot distribution…" on lines 158-159 of the revised manuscript.